# Online Active Learning with Surrogate Loss Functions

**Giulia DeSalvo**
Google Research
New York, NY 10011
giuliad@google.com

**Claudio Gentile**
Google Research
New York, NY 10011
cgentile@google.com

**Tobias Sommer Thune**
University of Copenhagen
Copenhagen, DK
tobias.thune@gmail.com

## Abstract

We derive a novel active learning algorithm in the streaming setting for binary classification tasks. The algorithm leverages weak labels to minimize the number of label requests, and trains a model to optimize a surrogate loss on a resulting set of labeled and weak-labeled points. Our algorithm jointly admits two crucial properties: theoretical guarantees in the general agnostic setting and a strong empirical performance. Our theoretical analysis shows that the algorithm attains favorable generalization and label complexity bounds, while our empirical study on 18 real-world datasets demonstrate that the algorithm outperforms standard baselines, including the Margin Algorithm, or Uncertainty Sampling, a high-performing active learning algorithm favored by practitioners.

## 1   Introduction

While unlabeled data is generated at increasing rates, supervised learning often requires large portions of this data to be annotated by human raters, which carry significant costs. Active learning seeks to significantly alleviate this problem through algorithms that attain comparable performance to passive learning (i.e., to full supervision) while using considerably fewer labels. Because of the ever increasing use of labeled data in modern machine learning approaches, this calls for the development of more and more effective active learning techniques.

In this paper, we are specifically interested in binary classification problems in the so-called streaming (or online) setting of active learning. In this setting, learning proceeds in a sequence of rounds, where in each round an unlabeled sample is received, and the learner decides on-the-fly whether or not to observe the associated binary label. Now, it is common practice, in active learning or machine learning in general, that even for binary classification problems, the loss used both for training and for evaluating statistical performance is not the zero-one loss, but some surrogate function thereof. This results in more tractable (e.g. convex) optimization problems in the training phase, but also in more calibrated metrics in the evaluation phase. For instance, we may be compelled to output estimates of class probabilities, say, the success rate of a given set of decisions, and thus deploy a maximum posterior probability estimator for such probabilities. Then, given a training set with labels $y \in \{\pm 1\}$ and if $\hat{p}(x)$ is an estimate of the conditional class probability $\mathbb{P}[y = 1 \,|\, x]$, a standard goal is to minimize the log-loss $\frac{1+y}{2} \log \frac{1}{\hat{p}(x)} + \frac{1-y}{2} \log \frac{1}{1-\hat{p}(x)}$. In these cases, the logistic link function $\hat{p}(x) = \frac{1}{1+e^{-h(x)}}$ is often adopted to model conditional class probabilities where $h$ is some (unknown) function in a given hypothesis class. In this and many other situations that arise in practice, it is thus advisable to resort to surrogate loss functions for both training and evaluation.

The online active learning literature contains a number of proposed algorithms that admit theoretical guarantees for both generalization error and label complexity, that is, the expected number of label requests. The theoretical analyses found in these works are either tailored to the zero-one loss or to general surrogate losses, with much more emphasis historically placed on the former. The papers that

35th Conference on Neural Information Processing Systems (NeurIPS 2021).

analyze zero-one loss introduce novel ideas tackling important theoretical questions in active learning, but the theory does not easily extend to general loss functions – see discussion in Henneke and Yang [2019]. Nevertheless, the algorithmic solutions we propose borrow several ideas from one such work, namely [Dasgupta et al., 2008]. In the agnostic setting, Dasgupta et al. [2008] developed an algorithm for the zero-one loss based on the disagreement between functions, that carefully constructs *pseudo*-labels, that is, weak labels generated by the algorithm itself. We call this algorithm DHM, after the name of the three authors.

For general surrogate losses, Beygelzimer et al. [2009] designed an algorithm based on constructing importance weighted predictors with theoretical guarantees in the agnostic setting. Cortes et al. [2019a,b, 2020] further developed these importance-weighted algorithms enhancing their guarantees and used them as building blocks for meta-algorithms. We refer to these algorithms as IWAL (Importance Weighted Active Learning).

On a different line of work, Henneke and Yang [2019] presents a theoretical analysis of an active learning algorithm that uses surrogate loss functions for binary classification. Among the theoretically-oriented contributions in active learning, this work appears to be the closest reference to our paper when it comes to motivations. Yet, it is fair to say that the authors make the very strong assumption that the Bayes optimal function for the surrogate loss at hand lies in the hypothesis class used for learning, and they explicitly state that this realizability condition cannot be relaxed without resorting to a significantly different approach. Moreover, unlike our work and that of IWAL-based algorithms, Henneke and Yang [2019] are only focusing on deriving bounds on the zero-one risk. And finally, their algorithm operates in a pool-based setting of active learning, where the whole set of unlabeled points is available to the learner, which is more flexible than the streaming setting analyzed here.

In practice, the Margin Algorithm, or Uncertainty Sampling, of Lewis and Gale [1994] attains a substantial performance improvement compared to passive learning as well as other active learning algorithms, and is thus often favored by practitioners (e.g. Schohn and Cohn [2000], Tong and Koller [2001], Brinker [2003], Culotta and McCallum [2005], Joshi et al. [2009], Mussmann and Liang [2018]). Moreover, the margin algorithm is flexible in that it can be used in both the pool and streaming setting. In the pool setting, several papers have recently tailored active learning algorithms to neural networks, always comparing their algorithm's empirical performance to that of the margin algorithm [Geifman and El-Yaniv, 2017, Gal et al., 2017, Savarese et al., 2018, Ducoffe and Precioso, 2018, Ash et al., 2020, Moon et al., 2020, Schroder and Niekler, 2020]. Overall, the margin algorithm is often found to be a competitive contender to this new collection of practical active learning algorithms. Additionally, two extensive empirical studies found that the margin algorithm outperforms many recent active learning algorithms [Yang and Loog, 2016, Chuang et al., 2019]. On the theoretical side, several authors developed active learning algorithms with theoretical guarantees for specific classes of functions (e.g., linear separators) based on sampling along the margin [Dasgupta et al., 2005, Balcan et al., 2007, Balcan and Long, 2013, Awasthi et al., 2014, 2015, Zhang, 2018, Zhang et al., 2020], but the guarantees of these margin-based algorithms only hold under strong assumptions on the realizability of the problem. Our goal is thus to derive an algorithm that admits theoretical guarantees without relying on realizability assumptions and that, at the same time, attains in practice a performance that is better than that of the margin algorithm in the streaming setting.

## 1.1 Contributions

In this paper, we introduce a novel active learning algorithm, called ALPS (Actively Learning over Pseudo-labels for Surrogate losses), in the streaming setting for binary classification tasks that trains a model by optimizing a surrogate loss on the joint set of labeled and self-constructed pseudo-labeled points (Section 3). Crucially, we prove that ALPS admits theoretical guarantees with respect to the surrogate loss in the general agnostic setting while at the same time surpassing the training performance of the margin algorithm as well as other baselines.

More concretely, we prove theoretical guarantees for ALPS in terms of both generalization and label complexity, under assumptions on the interplay between the function class and the data distribution which are not captured by traditional low noise assumptions often formulated in active learning (Section 4). We show that ALPS is able to leverage pseudo-labels in a principled manner even though the loss function of interest is not the zero-one loss, but a surrogate loss thereof. We then complement our theoretical findings with a thorough experimental investigation on 18 real-world

datasets using a class of neural networks, Multi-layer Perceptrons. We test our algorithm against the IWAL algorithm of Beygelzimer et al. [2009] since it admits theoretical guarantees for general loss functions, and against passive learning since we want to quantify the improvement over supervised methods. Additionally, we generate as competitors a pool of margin-based uncertainty samplers by varying their label request threshold, and show that on almost all datasets our algorithm outperforms the ex-post best among such algorithms (Section 5).

This paper follows the general research agenda of reducing the gap between theory and practice, which has been widening at a faster rate in the recent active learning literature. As outlined above, we try to bridge this disconnect by deriving an algorithm that not only works well in practice, but also admits solid theoretical guarantees. In doing so, we also introduce novel algorithmic ideas, which could independently lead to new directions of research.

## 1.2 Main Ideas

The main mechanism behind ALPS is to query the label of the current instance based on the disagreement between hypotheses and to construct pseudo-labels for the non-queried instances. The algorithm then trains a model on the joint set of labeled and pseudo-labeled data. The sample used for training will be unbiased with respect to marginal distribution on the instance space, but label noise will be introduced whenever the pseudo-label differs from the true (unrevealed) label. This noise affects both the generalization ability of the algorithm and its label complexity. Controlling this noise is not only crucial, but also quite challenging, for it relies on assessing the expected reliability of the pseudo-labels.

In order to control this noise, we introduce the novel idea of using requester functions: ALPS learns to query labels on instances with unreliable (that is, potentially noisy) pseudo-labels by using a requester function chosen to minimize an importance-weighted estimate of the noise. For the sake of our results, the requester functions are assumed to lie in some generic function class. One natural example of this class is that of *margin-based* functions, meaning functions that request along the margin of a given prediction model. In this case, our algorithm is related to the margin algorithm but, unlike the theoretical guarantees for the margin algorithm (e.g. Balcan et al. [2007]), we do not rely on specific distributional assumptions on the instance space. The requester functions are reminiscent of the abstention functions introduced in Cortes et al. [2016], but serve an entirely different purpose in active learning.

ALPS can be seen as an extension of the DHM algorithm [Dasgupta et al., 2008] to surrogate loss functions, which seeks to address the open question posed by Dasgupta et al. [2008] of whether there are active learning algorithms that only require solving tractable optimization problems in agnostic scenarios. At the same time, these two algorithms differ on how they treat the label noise introduced by pseudo-labeling. In DHM, the label noise can be easily bypassed by a simple property of the zero-one loss. In contrast, for general surrogate loss functions, this noise is unavoidable, and if not controlled as is done in the ALPS algorithm, the learned hypothesis can be arbitrarily far from the best-in-class.

## 2 Preliminaries and Notation

We consider an active learning framework in the streaming setting for binary classification. Learning proceeds in a sequence of rounds. In each round $n$, the learner receives from the environment an instance (or feature vector) $x_n$ from an instance (or feature) space $\mathcal{X}$. Based on what has been observed so far, the learner can then decide whether or not to request the true label $y_n$ associated with $x_n$. Label $y_n$ is assumed to lie in the output space $\mathcal{Y} = \{\pm 1\}$. The pairs $(x_1, y_1), (x_2, y_2), \dots$ are drawn i.i.d. from a joint distribution $D$ over $\mathcal{X} \times \mathcal{Y}$.

In order to evaluate the statistical performance of the learner, we consider any bounded surrogate loss function $\ell : \mathbb{R} \times \mathcal{Y} \to [0, B]$ that upper bounds the zero-one loss, and let $\ell_B(\cdot, \cdot) = \ell(\cdot, \cdot)/B$ be a $[0, 1]$-normalized version of this loss.

Let $H$ denote a hypothesis class of functions $h : \mathcal{X} \to \mathbb{R}$. For simplicity of exposition, we assume that $H$ is a finite class, but our analysis can be extended to classes with finite VC dimension by standard covering arguments. The true (expected) risk of hypothesis $h \in H$ is defined as $err(h) = \mathbb{E}_{(x,y) \sim D}[\ell(h(x), y)]$ and $h^* = \operatorname{argmin}_{h \in H} err(h)$ denotes the best-in-class hypothesis.

Moreover, we denote by $err(h, Z) = \frac{1}{n} \sum_{(x,y) \in Z} \ell(h(x), y)$ the empirical risk of $h$ over a training set $Z = \{(x_1, y_1), (x_2, y_2), \ldots, (x_n, y_n)\}$.

Given round $n \geq 1$, we denote by $T_n$ the set of labeled data points $\{(x_s, y_s)\}$ from rounds $s \in [n] := \{1, \ldots, n\}$ where the label was requested by the learning algorithm, and denote by $S_n$ its complement, that is, the set of labeled data points $\{(x_s, y_s)\}$ from rounds $s \in [n]$ where the label was not requested. Unlike $T_n$, the set $S_n$ is not fully known to the learner. On those rounds, the learner replaces the true unobserved label $y_s$ by pseudo-label $\hat{y}_s$, which is a suitable proxy to $y_s$ generated by the learner itself. We denote by $\hat{S}_n$ the set containing data points $\{(x_s, \hat{y}_s)\}$ from rounds $s \in [n]$ where the label is not requested, and the true label gets replaced by the corresponding pseudo-label.

For notational convenience (and unless otherwise specified), we set $\hat{y}_s = y_s$ in rounds $s$ where the true label is known to the learner. Then the following two short-hands are used for empirical risk up to round $n$ using ground-truth labels and the empirical risk up to round $n$ using pseudo-labels:

$$\underset{n}{err}(h) = err(h, S_n \cup T_n) = \frac{1}{n} \sum_{s=1}^{n} \ell(h(x_s), y_s); \quad \underset{n}{\widehat{err}}(h) = err(h, \hat{S}_n \cup T_n) = \frac{1}{n} \sum_{s=1}^{n} \ell(h(x_s), \hat{y}_s).$$

In the sequel, we will also need the following definition of consistency of hypothesis $h \in H$.

**Definition 1.** *We say that $h \in H$ is* consistent *with a labeled data point $(x, y) \in \mathcal{X} \times \mathcal{Y}$ if $\operatorname{sgn}(h(x)) = y$ and, by extension, that $h$ is* consistent *with a set of labeled data points $Z$ if it is consistent with all data points in $Z$. Two hypotheses $h, h' \in H$ are consistent with one another on a set of (unlabeled) points if $\operatorname{sgn}(h(x)) = \operatorname{sgn}(h'(x))$ for all $x$ in that set of points.*

At each round $n$, the learner is compelled to output a hypothesis $h_n \in H$, and we seek to bound its true risk, $err(h_n)$, in terms of the true risk, $err(h^*)$, of the best-in-class hypothesis $h^*$, as a function of the total number of streamed points $n$ thus far. In addition, we want a bound on the *label complexity* of the learner, that is, the number of labels requested by the learner during the course of training. This bound should be provably smaller than $n$, which is the label complexity of a passive learner observing all labels. Further ancillary definitions will be introduced in later sections.

## 3   The ALPS Algorithm

At a high level, ALPS either requests the label of an instance vector $x_n$ or assigns it a pseudo-label at each round $n$. The algorithm then selects a model $h_n$ that is consistent on the pseudo-labeled points, $\hat{S}_n$, and that minimizes the empirical estimate of the risk, $\widehat{err}_n(h)$, which is defined in terms of pseudo-labeled and labeled points, $\hat{S}_n \cup T_n$, processed thus far. See pseudo-code in Algorithm 1.

More concretely, the learned hypothesis at round $n$ is $h_n = \text{LEARN}(\hat{S}_n, \hat{S}_n \cup T_n)$ where:

**Definition 2.** *For sets of labeled data points $S$ and $Z$, $\text{LEARN}(S, Z)$ returns a hypothesis in $H$ that minimizes the empirical risk $\widehat{err}(h, Z)$ on the second argument $Z$, and is* consistent *with all labeled data points in the first argument $S$, where consistency is defined in Definition 1. LEARN raises a flag if no such hypothesis is found.*

In the LEARN procedure, we only need to consider hypotheses consistent with pseudo-labeled points, $\hat{S}_n$, because we can show that with high probability, the sign of the best-in-class, $h^* = \operatorname{argmin}_{h \in H} err(h)$, matches the sign of the pseudo-labeled points.

In order to decide whether to query the label of $x_n$, ALPS leverages the disagreements between hypotheses, $h_{+1}, h_{-1} \in H$ returned by the LEARN procedure, where the label of the current instance $x_n$ is assumed to be either +1 for learning hypothesis $h_{+1}$ and -1 for learning hypothesis $h_{-1}$. Ideally, the algorithm would use the empirical difference $err_n(h_{+1}) - err_n(h_{-1})$ based on ground-truth labels to assess this disagreement, but because not all labels at time $n$ have been disclosed to the algorithm, it instead uses the pseudo-labeled version $\widehat{err}_n(h_{+1}) - \widehat{err}_n(h_{-1})$, which is indeed observable. The algorithm compares this difference to a slack term, $\tilde{\Delta}_n$, derived from our theoretical analysis. See Appendix A for all slack term definitions. If the loss difference is smaller than the

---

[1]The sign of the pseudo-label $\hat{y}$ is determined by the condition $\widehat{err}_{n-1}(h_{-\hat{y}}) - \widehat{err}_{n-1}(h_{\hat{y}}) > \tilde{\Delta}_{n-1}$ and whether $h_{-\hat{y}}$ exist for $\hat{y} \in \pm 1$. E.g., $\hat{y} = +1$ if $\widehat{err}_{n-1}(h_{-1}) - \widehat{err}_{n-1}(h_{+1}) > \tilde{\Delta}_{n-1}$ or if $h_{-1}$ does not exist.

---

**Algorithm 1:** **A**ctively **L**earning over **P**seudo-labels for **S**urrogate losses (ALPS)

---

Input: LEARN$(S, Z)$, Hypothesis class $H$, requester class $R$, and slack terms $\tilde{\Delta}_n, \Delta'_n$.
$\hat{S}_0 = \emptyset; T_0 = \emptyset; F_1 = H \times R$;
**for** $n = 1, 2, \ldots, N$ **do**
    Receive feature vector $x_n$;
    For $\hat{y} = \pm 1$, let $h_{\hat{y}} = \text{LEARN}(\hat{S}_{n-1} \cup \{(x_n, \hat{y})\}, \hat{S}_{n-1} \cup \{(x_n, \hat{y})\} \cup T_{n-1})$;
    Define version space $F_n$ and bias $p_n$:

$$F_n = \{(h, r) \in F_{n-1} : \min_{(h', r') \in F_{n-1}} l_{n-1}(h', r') \geq l_{n-1}(h, r) - \Delta'_{n-1}\}$$

$$p_n = \max_{(h,r),(h',r') \in F_n} \max_{y \in \mathcal{Y}} \ell_B(y, h(x_n))\mathbb{I}\{r(x_n) \leq 0\} - \ell_B(y, h'(x_n))\mathbb{I}\{r'(x_n) \leq 0\}$$

    $Q_n \sim \text{Bernoulli}(p_n)$;
    **if** $\widehat{err}_{n-1}(h_{-\hat{y}}) - \widehat{err}_{n-1}(h_{\hat{y}}) > \tilde{\Delta}_{n-1}$(or if no such $h_{-\hat{y}}$ is found) for some $\hat{y} \in \{\pm 1\}$ AND $Q_n = 0$ **then**
        $r_n = \text{argmin}_{r \in R_n} \mathbb{E}[\mathbb{I}\{r(x) > 0\}]$, where

$$R_n = \{r : (h, r) \in F_n \text{ and } h \text{ consistent with } \hat{S}_{n-1} \cup \{(x_n, \hat{y})\}\}$$

        **if** $r_n(x_n) > 0$ **then**
            Request $y_n$ and update $\hat{S}_n = \hat{S}_{n-1}, T_n = T_{n-1} \cup \{(x_n, y_n)\}$;
        **else**
            Do not request $y_n$ and use[1]pseudo-label $\hat{y}$ to update $\hat{S}_n = \hat{S}_{n-1} \cup \{(x_n, \hat{y})\}, T_n = T_{n-1}$;
        **end if**
    **else**
        Request $y_n$ and update $\hat{S}_n = \hat{S}_{n-1}, T_n = T_{n-1} \cup \{(x_n, y_n)\}$;
    **end if**
    $h_n = \text{LEARN}(\hat{S}_n, \hat{S}_n \cup T_n)$;
**end for**
Output $h_N = \text{LEARN}(\hat{S}_N, \hat{S}_N \cup T_N)$;

---

slack, then the label is requested because the sign of best-in-class $h^*$ cannot be inferred. Otherwise, a pseudo-label is constructed. In more detail, the sign of the pseudo-label $\hat{y}$ is determined by the condition $\widehat{err}_{n-1}(h_{-\hat{y}}) - \widehat{err}_{n-1}(h_{\hat{y}}) > \tilde{\Delta}_{n-1}$ and whether $h_{-\hat{y}}$ exist for $\hat{y} \in \pm 1$. Note that by this construction, the LEARN procedure can always return a consistent hypothesis on all pseudo-labeled points.

To understand the next steps in the ALPS algorithm, we relate the ground-truth empirical difference $err_n(h) - err_n(h')$ to its the pseudo-labeled counterpart $\widehat{err}_n(h) - \widehat{err}_n(h')$. To this effect, we define the difference of *difference of losses* $A_n(h, h')$ between two hypotheses $h, h' \in H$ at time $n$ as

$$A_n(h, h') := err_n(h) - err_n(h') - \left(\widehat{err}_n(h) - \widehat{err}_n(h')\right).$$

This idea is also adopted by Dasgupta et al. [2008] in their analysis of the DHM algorithm. Yet, because they only deal with zero-one loss, in their case $A_n(h, h') = 0$ for consistent $h, h'$ even when the true label does not match the pseudo-label. This easily follows from the fact that if $\text{sgn}(h)$ and $\text{sgn}(h')$ agree on instance $x$, then[2] $\mathbb{I}\{\text{sgn}(h(x)) \neq y\} - \mathbb{I}\{\text{sgn}(h'(x)) \neq y\} = \mathbb{I}\{\text{sgn}(h(x)) \neq \hat{y}\} - \mathbb{I}\{\text{sgn}(h'(x)) \neq \hat{y}\}$, irregardless of whether $y = \hat{y}$ or not.

In the surrogate loss setting analyzed here, whenever the pseudo-labels do not match their corresponding true labels, the term $A_n(h, h')$ is typically non-zero even for consistent $h, h'$. As a consequence, the magnitude of $A_n(h, h')$ must be carefully controlled since, otherwise, it would break both generalization and label complexity bounds as well as the guarantee that the best-in-class $h^*$ is consistent with $\hat{S}_n$. In the sequel, we call $A_n$ the *noise term*, as it captures the noise introduced during training due to the discrepancy between pseudo-labels and true labels.

---

[2]Throughout, $\mathbb{I}\{\cdot\}$ is the indicator function of its argument.

To control the noise term, we introduce the idea of using *requester* functions. Specifically, we consider a set $R$ of functions $r: \mathcal{X} \to \mathbb{R}$, where $r(x) > 0$ means that the label of $x$ is requested by $r$, and otherwise it is not requested. At each round $n$, the algorithm picks a requester function $r_n$ out of the set $R$, and the algorithm's final condition of whether or not to request the label is decided by the sign of $r_n(x_n)$. The requester function $r_n$ thus works as the final gatekeeper of whether to use the pseudo-label.

Notice that by definition the noise term $A_n$ is non-zero when the true labels do not match the pseudo-labels, $y_s \neq \hat{y}_s$ for some $s \in [n]$. By construction, since the final condition of whether to use a pseudo-label in round $s$ is determined by $r_s$, it follows that $\mathbb{I}\{y_s \neq \hat{y}_s\} = \mathbb{I}\{y_s \neq \hat{y}_s\}\mathbb{I}\{r_s(x_s) \leq 0\}$. In addition, for any $h$ consistent with $\hat{S}_s$, it holds that $\hat{y}_s = \mathrm{sgn}(h(x_s))$, and since the surrogate loss $\ell$ upper bounds the zero-one loss, we have that for any $h$ consistent with $\hat{S}_s$,

$$\mathbb{I}\{y_s \neq \hat{y}_s\}\mathbb{I}\{r_s(x_s) \leq 0\} \leq \ell(h(x_s), y_s)\mathbb{I}\{r_s(x_s) \leq 0\} = B\ell_B(h(x_s), y_s)\mathbb{I}\{r_s(x_s) \leq 0\}. \quad (1)$$

The above can thus be used as an upper bound of the noise.

To estimate this upper bound of $A_n$, the algorithm constructs an unbiased importance-weighted estimate, $l_n(h, r)$, of $\mathbb{E}[\ell_B(h(x), y)\mathbb{I}\{r(x) \leq 0\}]$ as follows:

$$l_n(h, r) = \frac{1}{n} \sum_{s=1}^{n} \frac{Q_s}{p_s(x_s)} \ell_B(h(x_s), y_s)\mathbb{I}\{r(x_s) \leq 0\},$$

where $Q_s$ is a Bernoulli random variable with bias $p_s(x_s)$ and where $p_s(\cdot)$ is a (random) function only depending on the past history $\{(x_{s'}, y_{s'}), Q_{s'}\}_{s'=1}^{s-1}$. The algorithm will request the label $y_s$ whenever $Q_s = 1$ to construct this unbiased estimate. In order not to request the label ($Q_s = 1$) too frequently, $p_s(\cdot)$ is based on a shrinking version space $F_n$ that contains only the pairs $(h, r)$ whose empirical estimate $l_n(h, r)$ is close to the smallest one.

Given these estimates of $\mathbb{E}[\ell_B(h(x), y)\mathbb{I}\{r(x) \leq 0\}]$, the algorithm chooses a requester function $r_n$ with the smallest request rate $\mathbb{E}[\mathbb{I}\{r(x) > 0\}]$ from a set of requester functions in the version space $F_n$. In this way, the algorithm ensures that the noise term $A_n$ is kept small. This is due to the fact that the version space $F_n$ contains pairs of function $(h, r)$ that are provably close to those minimizing $\mathbb{E}[\ell_B(h(x), y)\mathbb{I}\{r(x) \leq 0\}]$, combined with the fact that $\ell_B(h(x_n), y_n)\mathbb{I}\{r_n(x_n) \leq 0\}$ is an upper bound on the noise at round $n$ via (1).

Putting the above together, ALPS will request the label of $x_n$ whenever one of the following three conditions hold: 1) the loss differences of $h_+$ and $h_-$ is smaller than the slack term, in which case the revealed label is used as we cannot infer the sign of $h^*$ and cannot construct a pseudo-label 2) $Q_n = 1$, in which case the revealed label is used for the empirical estimates $l_n(h, r)$, 3) $r_n(x_n) > 0$, in which case the revealed label is used to control the noise term, $A_n$, introduced by pseudo-labeling. In all cases, the revealed labels are used in LEARN to find the best model at each round $n$.

Notice that, for the sake of argument, we have assumed in the above description to have access to a large unlabeled pool of examples $x$ so as to construct a convenient set of requester functions, that is, to estimate $\mathbb{E}[\mathbb{I}\{r(x) > 0\}]$ accurately. Yet, our algorithm and its associated analysis can be generalized straightforwardly to the case when such a pool is available only in a streaming way. Moreover, in our experiments in Section 5, we do not use an unlabeled pool to generate requester functions or to estimate the requesting rate. Instead, we define the set of requester functions to be margin-based functions with varying thresholds, and then estimate the requesting rate based on the data processed thus far. See Section 5 for details.

## 4 Theoretical Guarantees

We now present generalization and label complexity guarantees of our algorithm. These bounds will be in terms of the best-in-class hypothesis, $h^*$, whose risk $err(h^*)$ will be denoted by $\nu$. For these guarantees, we consider a class $R$ of requester functions that satisfies the following:

**Assumption 1.** *There exists $r^* \in R$ and a constant $C \geq 0$ such that $\mathbb{E}[\ell(y, h^*(x))\mathbb{I}\{r^*(x) \leq 0\}] = 0$ and $\mathbb{E}[\mathbb{I}\{r^*(x) > 0\}] \leq C\nu$.*

If given access to a large unlabeled pool of data, we can always construct a function $r$ such that $\mathbb{E}[\ell(y, h(x))\mathbb{I}\{r(x) \leq 0\}] = 0$ for any $h$, since $r$ can always be augmented in such a way to request

more frequently (that is, making $r(x) > 0$ for most $x$). Moreover below, we will further relax the first constraint to $\mathbb{E}[\ell(y, h(x))\mathbb{I}\{r(x) \leq 0\}] \leq \epsilon$ for some $\epsilon \geq 0$. However, unless more information about the structure of the problem is known, proving that there exists a function $r^*$ in the set $R$ such that $\mathbb{E}[\mathbb{I}\{r^*(x) > 0\}] \leq C\nu$ holds with a favorable (i.e., reasonably small) value of $C$ is more difficult. Nevertheless, unfavorable $C$ values, just like unfavorable disagreement coefficients, only impact the label complexity bound and not the generalization guarantee. After the theorems below, we present several natural examples of favorable $C$ values.

Note that Assumumption 1 does not dictate conditions on the true conditional class probability, $\eta(x) = \mathbb{P}[y = 1 \,|\, x]$. Hence, we are not relying on low-noise assumptions often adopted in active learning in realizable scenarios. Specifically, our assumption cannot be viewed as a surrogate loss counterpart to the Tsybakov [Mammen and Tsybakov, 1999, Tsybakov, 2004] low noise condition in the realizable setting [Castro and Nowak, 2008, Hanneke, 2011, Koltchinskii, 2010, Dekel et al., 2012] nor as an incarnation of the Bernstein condition, often invoked in statistical learning settings to achieve fast rates in both passive and active learning [Massart and Nedelec, 2006, Bartlett et al., 2006, Koltchinskii, 2006, Van Erven et al., 2015, Henneke and Yang, 2019].

In general, the richer the $R$ class is, the easier Assumption 1 is satisfied with a small $C$. At the same time, since ALPS is learning over $H$ and $R$ simultaneously, the complexity of $R$ affects both generalization bound and label complexity, as per usual in learning guarantees. Below, we first present the generalization bound of the ALPS algorithm. Here and in Theorem 2, the $\tilde{O}$ notation is hiding logarithmic factors in $n$, $1/\delta$, and $|H \times R|$.

**Theorem 1.** *Under Assumption 1, for any $\delta > 0$, with probability at least $1 - \delta$, for any $n > 0$,*

$$err(h_n) \leq \nu + \tilde{O}\Big(\sqrt{\tfrac{\nu}{n}} + \tfrac{1}{n}\Big),$$

*where $h_n$ is the hypothesis computed by ALPS in round $n$.*

This guarantee states that the risk of the hypothesis returned by the algorithm converges to the best-in-class hypothesis at a rate that matches that of standard supervised learning, despite the fact that ALPS uses less labeled points. The analysis used to derive this theorem proves that the noise introduced by the pseudo-labels is, in fact, controlled by the learned requester function so that the above bound can be attained. In passing, we observe that if the loss $\ell$ is convex and the Bayes optimal hypothesis is in the class $H$, then by using standard consistency techniques [Zhang, 2004, Bartlett et al., 2006] in conjunction with Theorem 1 above, we could also attain a bound on the excess risk of the zero-one loss.

The next theorem shows a bound on the expected number of labels the algorithm requests. This bound will be in terms of a notion of disagreement coefficient derived from the one in [Hanneke, 2007]. Define a metric $\rho$ on the space of hypotheses $H$ as $\rho(h, h') := \mathbb{E}_{(x,y)\sim D}\big[|\ell(h(x), y) - \ell(h(x), y)|\big]$. Using this metric, we consider the $\gamma$-ball around the best-in-class: $B_\gamma(h^*) = \{h : \rho(h, h^*) \leq \gamma\}$. Then, let $\theta := \sup_{\gamma > 0} \{\mathbb{P}_x[\exists h \in B_\gamma(h^*) : \text{sgn}(h(x)) \neq \text{sgn}(h^*(x))]/\gamma\}$ be the disagreement coefficient of $H$ (for the given distribution $D$ over $\mathcal{X} \times \mathcal{Y}$).

**Theorem 2.** *Under Assumption 1, for any $\delta > 0$, with probability $1 - \delta$, the label complexity of the ALPS algorithm at time $n > 0$ is bounded as $\tilde{O}\left(n\nu(\theta + C)\right)$.*

Disregarding constants $\theta$ and $C$, the above bound is $\tilde{O}(n\nu)$. This label complexity guarantee of ALPS in conjunction with its generalization bound matches the known lower bound $\Omega(n\nu)$ from Beygelzimer et al. [2009] up to constants and is therefore optimal.

In Assumption 1, we considered the case when $\mathbb{E}[\ell(y, h^*(x))\mathbb{I}\{r^*(x) \leq 0\}] = 0$ which is easy to satisfy if we have a rich enough class $R$ with functions that request more frequently. Nevertheless, we can relax this assumption if desired as follows:

**Assumption 2.** *There exists $r^* \in R$ and constants $C \geq 0$ and $\epsilon \geq 0$ such that $\mathbb{E}[\ell(y, h^*(x))\mathbb{I}\{r^*(x) \leq 0\}] \leq \epsilon$ and $\mathbb{E}[\mathbb{I}\{r^*(x) > 0\}] \leq C\nu$.*

Under this assumption, the label complexity guarantee in Theorem 2 remains unchanged while an $\epsilon$ term is added to the generalization bound in Theorem 1 when ALPS is run with slack $\tilde{\Delta}_{n-1} + \epsilon$. By definition, $\epsilon$ is at most $\nu$ and if $\epsilon \ll \nu$, then this term has basically no effect on generalization.

There are several natural instances where the condition stated in Assumption 1 and Assumption 2 are satisfied with favorable $C$ values. Consider for example a scenario where along the margin of the

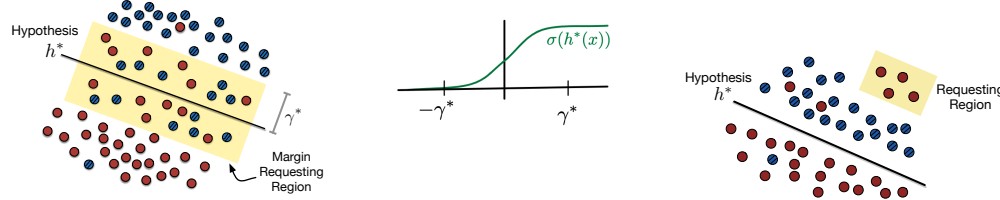

Figure 1: Left: In yellow is a margin requesting region $\{x \in \mathcal{X} : |h^*(x)| < \gamma^*\}$, where the hypothesis $h^*$, depicted by the black line, admits a conditional error of, say, 1/2. Middle: A tapered sigmoid function that equals 1 or 0 outside the $[-\gamma^*, \gamma^*]$ interval. Right: An example of a requesting region (in yellow) that is not margin-based and where Assumption 2 holds with $C < 1$.

best-in-class hypothesis, there are hard-to-classify examples, while further away the best-in-class hypothesis mostly classifies the examples correctly. See Figure 1 for an illustration where there is such a margin $\gamma^*$ around the best-in-class $h^*$. This example is often used to explain why the popular margin algorithm works since this algorithm queries the label of points close to the classification surface. In our case, letting $r^*(x) = \gamma^* - |h^*(x)|$, it holds that $\mathbb{E}[\ell(y, h^*(x))\mathbb{I}\{|h^*(x)| \geq \gamma^*\}] \leq \epsilon$ for a small $\epsilon$ and $\mathbb{E}[\mathbb{I}\{|h^*(x)| < \gamma^*\}] = \frac{\nu}{\mathbb{E}[\ell(y,h^*(x)) \,|\, |h^*(x)|<\gamma^*]}$. If the expected loss of $h^*$ conditioned on the margin region is high, say $\geq 1/2$, then it follows that $\mathbb{E}[\mathbb{I}\{r^*(x) > 0\}] = \mathbb{E}[\mathbb{I}\{|h^*(x)| < \gamma^*\}] \leq 2\nu$, so that Assumption 2 is satisfied with $C = 2$. Specifically, in Figure 1, it holds that $C = 2$ and $\epsilon = 2/50$.

When $h^*$ performs well outside the margin, then Assumption 2 holds with a small $\epsilon$ while if it correctly classifies all points outside the margin, then $\epsilon = 0$ and Assumption 1 holds. The $\epsilon = 0$ case is easily satisfied by generalized linear models, $\mathbb{P}[y = 1|x] = \sigma(h^*(x))$, where $\sigma$ is a tapered sigmoid function. Note that Assumption 1 is not placing specific restrictions on the marginal distribution over $\mathcal{X}$, other than $\mathbb{E}[\mathbb{I}\{|h^*(x)| < \gamma^*\}] = 2\nu$. This does not imply conditions on $\eta(x)$ when $x$ falls in the region $|h^*(x)| < \gamma^*$ and hence we are not relying on realizability assumptions.

With any given class of hypotheses $H$, one can always associate the margin-based requester function class $R = \{r : r(x) = \gamma - |h(x)|, \gamma \geq 0, h \in H\}$. In this case, ALPS can be seen as trying to simultaneously approximate $h^*$ and the best threshold $\gamma^*$. The resulting algorithm turns out to be related to margin-based approaches since ALPS will seek to query the label of these more difficult points along the margin, but ALPS finds this optimal pair by using a different approach.

At the same time, we would like to stress that, in our framework, the function class $R$ need not be restricted to margin-based requesters: if there exists a small region, $\{x \in \mathcal{X} : r^*(x) > 0\}$, for some function $r^* : \mathcal{X} \to \mathbb{R}$ such that most of the loss incurred by the best-in-class $h^*$ is coming from this region, and $R$ is rich enough to contain such function $r^*$, then Assumption 1 and Assumption 2 hold. For an illustration of this situation, see Figure 1 on the right.

### 4.1 Comparisons to IWAL and DHM

The IWAL algorithm of Beygelzimer et al. [2009] constructs importance weighted estimates of the expected loss and uses them to select the prediction function. In contrast, ALPS leverages importance weighted estimators to select requester functions that minimize the noise term. Thus, even though both algorithms construct importance weighted estimators, they serve entirely different purposes. Both IWAL and ALPS attain generalization and label complexity bounds that are effectively of the same order, but our empirical results show that IWAL considerably underperforms compared to other active learning algorithm on all 18 datasets we tested. These empirical results are consistent with other authors' findings (e.g., Figure 2 in Cortes et al. [2019b] and Figure 3 in Cortes et al. [2020]).

Even though ALPS can be seen as a generalization of DHM [Dasgupta et al., 2008] to surrogate loss functions, there are several key differences. The noise introduced by the pseudo-labels does not affect DHM since by definition $A_n(h, h') = 0$ for the zero-one loss. Thus, DHM does not resort to requester functions or other techniques to deal with unreliable pseudo-labels. Moreover, unlike in DHM, the second argument in the LEARN$(\cdot, \cdot)$ subroutine over which ALPS minimizes error is $\hat{S}_n \cup T_n$, not just $T_n$. For the zero-one loss, minimizing the error over $\hat{S}_n \cup T_n$ or only $T_n$ is equivalent since hypothesis consistent on $\hat{S}_n$ admit zero loss over the points in $\hat{S}_n$. Note that ALPS's pseudo-code

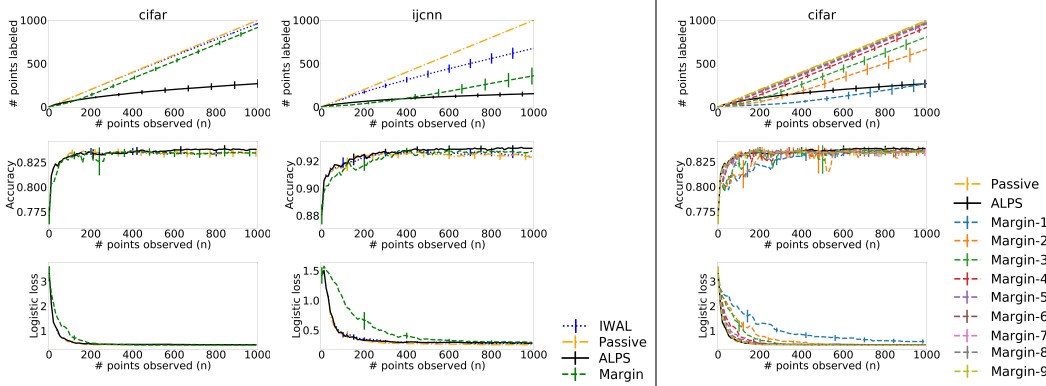

Figure 2: The plots on the left show the performance of ALPS, IWAL, the ex-post best margin algorithm, and passive learning with 4-layer networks for `cifar` and with 3-layer networks for `ijcnn`. For the 4-layer networks for `cifar`, the plot on the right shows the margin algorithms with thresholds in $\Gamma_m$ ordered from smallest to largest threshold in $\Gamma_m$ where Margin-1 corresponds to the smallest threshold value in $\Gamma_m$, Margin-2 to the second smallest, etc. As a function of the number of observed points (or rounds) $n$, the plots show the mean over the 50 repetitions and twice its standard error of the number of requested labels as well as the accuracy and logistic loss on the test set of the model returned by each algorithm.

reduces to DHM's when $\ell$ is the zero-one loss and $R$ is set to include only the never-requesting function, that is, $r(x) = 0$ for all $x$, in which case the importance-weighted estimates $l_n(h, r)$ are not constructed. We then attain the same guarantees as DHM's without resorting to Assumption 1 or Assumption 2.

## 5 Empirical Results

In this section, we present our experimental results in the streaming active learning setting that test the ALPS algorithm, the IWAL algorithm, the margin algorithm (or uncertainty sampling), and a passive learning algorithm, which requests the label of all points and finds the hypothesis that attains the smallest empirical loss. We compare to IWAL since it is a principled algorithm in the general agnostic scenario with strong theoretical guarantees. We test the margin algorithm since even though its theoretical guarantees do not hold in general, it admits a strong performance in practice. We run a passive learning algorithm to demonstrate the benefit of ALPS over standard supervised learning.

We tested these algorithms on 18 publicly available datasets where we used the logistic loss as the surrogate loss $\ell$ and used feedforward artificial neural networks as our model class. Specifically, we used the Multi-layer Perceptron algorithm in the `scikit-learn` library and ran two diverse network architectures, one with 3-layers and another with 4-layers. For each type of neural network, we constructed a diverse finite set of hypotheses by pre-training on small random subsets while varying the $l_2$ regularization parameter and initial weights. For each experiment, we randomly shuffled the dataset, generated the finite hypothesis set, and ran all the algorithms. We repeated the experiment 50 times and averaged the results. See Appendix D for details on the datasets and model class settings.

Recall that the margin algorithm in the streaming setting first selects the hypothesis $h \in H$ with the smallest empirical loss and then requests the label of a point $x$ if $|\mathbb{P}_h[y = +1|x] - \mathbb{P}_h[y = -1|x]| \leq \gamma$, where $\mathbb{P}_h[y = \pm 1|x]$ is the model $h$'s (estimated) conditional probability of labeling the given point $x$ either $+1$ or $-1$. We ran nine instances of the margin algorithm for all thresholds $\gamma \in \Gamma_m$ where the set of thresholds $\Gamma_m$ was chosen as a result of a tuning procedure that consisted of a standard grid search and zooming in. For ALPS, the requester class $R$ is also defined by margin-based functions as $r(x) = \mathbb{I}\{|\mathbb{P}_h[y = +1|x] - \mathbb{P}_h[y = -1|x]| \leq \gamma\}$ for each hypothesis $h \in H$ and each $\gamma \in \Gamma_r$. Note that we run ALPS with $\epsilon = 0$ for this class $R$. Unlike $\Gamma_m$, the set $\Gamma_r$ was not tuned since ALPS learns the best threshold in $\Gamma_r$. For details on $\Gamma_m$ and $\Gamma_r$, please see Appendix D.

To evaluate the algorithm performance, both accuracy and logistic loss are of interest in different applications as explained in the introduction. Thus, Figure 2 shows the accuracy and logistic loss on

the test set of the model returned by each algorithm at each round $n$. Figure 2 also plots the number of points labeled by each algorithm as a function of the number of unlabeled points (or rounds) $n$.

Notice, in particular, that for the passive learning algorithm, since the data is streamed in a random order the value of the curves at time step $n$ can be interpreted as running the algorithm up until we gather $n$ queries on randomly drawn points. So, for instance, at round $n = 200$ of the passive curve (orange line) for the `ijcnn` dataset, we effectively randomly sample and reveal the label of 200 points out of the training set, choose a model that corresponds to an empirical risk minimizer over these 200 points, and evaluate this model's accuracy on the test set. The passive learning curve is then used as a baseline comparator of the active learning algorithms. That is, the best performing active learning algorithm is the one that attains an accuracy and logistic loss curve that is on par or better than that of passive learning while requesting the fewest number of labels.

Figure 2 on the right shows that the accuracy and logistic loss curves of the margin algorithm varies with the threshold. In almost all datasets, there exists thresholds whose corresponding margin algorithm admits an accuracy curve that is below that of passive learning and as the threshold increases, the accuracy curve improves eventually matching that of passive learning. This indicates that a good set of threshold values was tested, since we seek an algorithm that requests the fewest number of points (i.e. having a small threshold) with an accuracy curve on par to that of passive learning. We then pick the ex-post best margin algorithm with smallest threshold that admits an learning curve area with respect to accuracy that is within one standard error of the passive learning curve area. If no algorithm admits such a curve, we pick the margin algorithm with the largest learning curve area. Choosing the ex-post best margin algorithm in this way and tuning of $\Gamma_m$ gives the margin algorithm an unfair advantage, since both of these processes use the learning curves based on the revealed labels. Nevertheless, it provides an upper bound on the best performance of these uncertainty samplers.

The left-most plots of Figure 2 show that, for both `cifar` and `ijcnn` datasets, the ALPS algorithm performs better than all baseline methods since out of the algorithms that attain the accuracy and logistic loss curve close to that of passive learning, it requests the fewest number of points. In Appendix D, we report these figures for all 18 datasets.

Overall for the 3-layer networks, our results show that the ALPS algorithm outperforms the ex-post best margin algorithm on all but one dataset. Similarly, for the 4-layer networks, ALPS performs better than the margin algorithm on all but two dataset. Thus, despite the advantage given by the ex-post tuning of the margin algorithm, ALPS attains a better performance. Moreover, ALPS outperforms both passive learning and IWAL on all datasets. On average across all datasets, ALPS requests only a mere 28% of the processed points. Since ALPS was run with $\epsilon = 0$, these experiments suggest that Assumption 1 holding with favorable $C$ values often happens in practice or that Assumption 1 is simply an artifact of our analysis.

Additionally, the ex-post best margin algorithm outperforms IWAL on the majority of the datasets, which is consistent with previous studies [Cortes et al., 2020]. IWAL attains a higher variance across the trials for some datasets, and overall it performs better than passive learning. In general, the same empirical conclusions about the relative performance of the active learning algorithms hold for logistic loss and accuracy curves for all algorithms except for the margin algorithm. On a few datasets, the margin algorithm attains a logistic loss curve that is worse than that of passive learning while admitting an accuracy curve that is on par to that of passive learning (e.g., `ijcnn`).

## 6   Conclusion

We designed an active learning algorithm, called ALPS, for general loss functions that learns to leverage pseudo-labels by using requester functions in order to train a model over a joint set of pseudo-label and labeled points. ALPS operates in the general agnostic setting; its model is guaranteed to converge to the best-in-class prediction model at the same rate as passive learning, while achieving favorable label complexity guarantees under a mild assumption on the class of requester functions. Our comprehensive empirical study on 18 datasets shows that ALPS outperforms relevant baselines, including the margin algorithm often used in practice. As a next step, we will investigate how to extend our algorithm and its associated analysis to multi-class classification setting.

**Acknowledgments.** Most of this research was done while the Tobias Sommer Thune was visiting Google Research in New York. Tobias Sommer Thune further acknowledges partial support by the Independent Research Fund Denmark, grant number 9040-00361B. We thank the NeurIPS anonymous reviewers whose comments helped us improve the presentation of this paper.

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
