# A Proofs of the Generalization Bound

In this appendix, we prove the generalization guarantees of the hypothesis returned by the ALPS algorithm. We first derive necessary concentration inequalities, then prove consistency guarantees, and show how to control the noise $A_n$. Throughout, we will be making use of some technical lemmas in Appendix C. For simplicity, we define a *disagreement-based request* at time $n$ to be when $h_-, h_+$ exist and $\widehat{err}_{n-1}(h_{-\hat{y}}) - \widehat{err}_{n-1}(h_{\hat{y}}) \leq \tilde{\Delta}_{n-1}$ for $\hat{y} \in \{\pm 1\}$.

## A.1 Concentration Inequalities

In this section, we provide a series of concentration inequalities used throughout. We consider finite classes, but our analysis can straightforwardly be generalized to VC classes via standard covering arguments. We will need the following definition:

$$\Lambda_n(|\mathcal{H}|) = 2\sqrt{B}\sqrt{\frac{4\log(n(n+1)|\mathcal{H}|/\delta)}{n}} \left( \frac{1}{2} + \sqrt{1 + \frac{1}{2}\log\left(\frac{1}{2}\sqrt{\frac{n}{4\log(n(n+1)|\mathcal{H}|/\delta)}}\right)} \right),$$

where $\mathcal{H}$ is a hypothesis space of cardinality $|\mathcal{H}|$.

**Lemma 1.** *Consider a hypothesis space $\mathcal{H}$ of cardinality $|\mathcal{H}|$ and a loss function $g : \mathbb{R} \times \mathcal{Y} \to [0, B]$. Let $Z_n$ be an i.i.d sample of size $n$ from the underlying distribution and let $\mathbb{E}_{Z_n}[g]$ be the empirical mean of $g$ over the set $Z_n$. Then, for any $\delta > 0$, with probability at least $1 - \delta$, it holds for all $n$ and all $h \in \mathcal{H}$:*

$$- \min\left( \Lambda_n(|\mathcal{H}|)\sqrt{\mathbb{E}[g]}, \Lambda_n(|\mathcal{H}|)^2 + \Lambda_n(|\mathcal{H}|)\sqrt{\mathbb{E}_{Z_n}[g]} \right)$$

$$\leq \mathbb{E}_{Z_n}[g] - \mathbb{E}[g]$$

$$\leq \min\left( \Lambda_n(|\mathcal{H}|)\sqrt{\mathbb{E}_{Z_n}[g]}, \Lambda_n(|\mathcal{H}|)^2 + \Lambda_n(|\mathcal{H}|)\sqrt{\mathbb{E}[g]} \right).$$

*Proof.* The lemma follows from a direct application of Corollary 14 of [Cortes et al., 2013]. Note that the second moments fulfill $\mathbb{E}[g^2] \leq B\,\mathbb{E}[g]$ as the loss $g$ is assumed to be bounded and non-negative. For finite class $\mathcal{H}$, the shattering coefficient of set $Z_n$, denoted by $\mathbb{S}_q(Z_n)$, is bounded by $\mathbb{S}_q(Z_n) \leq n|\mathcal{H}|$. Replacing $\delta$ by $\delta/n(n+1)$ while taking a union bound over all $n$, the following inequalities hold for all $n$ and all $h \in \mathcal{H}$, for any $\delta > 0$, with probability at least $1 - \delta$:

$$\mathbb{E}[g] \leq \mathbb{E}_{Z_n}[g] + \Lambda_n(|\mathcal{H}|)\sqrt{\mathbb{E}[g]},$$

$$\mathbb{E}_{Z_n}[g] \leq \mathbb{E}[g] + \Lambda_n(|\mathcal{H}|)\sqrt{\mathbb{E}_{Z_n}[g]}.$$

The bound follows follows directly by applying Lemma 7 (Appendix C) to inequalities above. $\square$

Next, we derive a guarantee based on the loss differences, which will dictate the form of the final generalization guarantee of the ALPS algorithm. To do so, we define the following functions:

$$g_{h,h'}^+(x, y) = \left(\ell(h(x), y) - \ell(h'(x), y)\right)\mathbb{I}\{\ell(h(x), y)) > \ell(h'(x), y)\},$$

$$g_{h,h'}^-(x, y) = \left(\ell(h'(x), y) - \ell(h(x), y)\right)\mathbb{I}\{\ell(h(x), y)) < \ell(h'(x), y)\}.$$

so that the difference of $g_{h,h'}^+$ and $g_{h,h'}^-$ equals to the loss difference of $h$ and $h'$, both empirically and in expectation:

$$\mathbb{E}_Z[g_{h,h'}^+] - \mathbb{E}_Z[g_{h,h'}^-] = err(h, Z) - err(h', Z),$$

$$\mathbb{E}[g_{h,h'}^+] - \mathbb{E}[g_{h,h'}^-] = err(h) - err(h').$$

Both $g^+$ and $g^-$ can be seen as bounded loss functions of a hypothesis $(h, h') \in H^2$ and as such we can apply Lemma 1 to the the space $\mathcal{H} = H^2$ of cardinality $|H|^2$, where we recall that $H$ is the hypothesis class used in the algorithm.

For below, we let $\tilde{A}_n(h, h') = \frac{1}{n}\sum_{s=1}^n |\ell(h(x_s), y_s) - \ell(h'(x_s), y_s) - (\ell(h(x_s), \hat{y}_s) - \ell(h'(x_s), \hat{y}_s))|$, which is equivalent to the definition of $A_n(h, h')$ except for absolute values and will be dealt with in a similar way.

**Corollary 1.** *For any $\delta > 0$, with probability at least $1 - \delta$, for any $h, h' \in H$ and any $n$,*

$$\widehat{err}_n(h) - \widehat{err}_n(h') \le err(h) - err(h') - A_n(h, h') + \Lambda_n(|H|^2)^2 + 2\Lambda_n(|H|^2)\sqrt{\tilde{A}_n(h, h') + \widehat{err}_n(h) + \widehat{err}_n(h')}.$$

*Proof.* First, we recall the definitions of $A_n(h, h')$:

$$A_n(h, h') = err_n(h) - err_n(h') - \left[\widehat{err}_n(h) - \widehat{err}_n(h')\right].$$

Letting $Z$ be the full data set $S_n \cup T_n$ with true, unknown labels, it follows that $\mathbb{E}_Z[g^+_{h,h'}] - \mathbb{E}_Z[g^-_{h,h'}] = err(h, S_n \cup T_n) - err(h', S_n \cup T_n) = err_n(h) - err_n(h')$. Using this fact and rewriting $A_n$, it holds that

$$\widehat{err}_n(h) - \widehat{err}_n(h') = \mathbb{E}_Z[g^+_{h,h'}] - \mathbb{E}_Z[g^-_{h,h'}] - A_n(h, h').$$

Applying Lemma 1 to $g^+_{h,h'}$ and $g^-_{h,h'}$, for $h, h' \in H$,

$$\widehat{err}_n(h) - \widehat{err}_n(h') \le \mathbb{E}[g^+_{h,h'}] - \mathbb{E}[g^-_{h,h'}] - A_n(h, h') + \Lambda_n(|H|^2)^2$$
$$+ \Lambda_n(|H|^2)\sqrt{\mathbb{E}_Z[g^+_{h,h'}]} + \Lambda_n(|H|^2)\sqrt{\mathbb{E}_Z[g^-_{h,h'}]}.$$

Next, we can bound the expectations under the square roots :

$$\mathbb{E}_Z[g^+] \le \frac{1}{n}\sum_{s=1}^{n} |\ell(h(x_s), y_s) - \ell(h'(x_s), y_s)|$$

$$= \frac{1}{n}\sum_{s=1}^{n} |\ell(h(x_s), y_s) - \ell(h(x_s), \hat{y}_s) + \ell(h(x_s), \hat{y}_s) - \ell(h'(x_s), \hat{y}_s) + \ell(h'(x_s), \hat{y}_s) - \ell(h'(x_s), y_s)|$$

$$\le \frac{1}{n}\sum_{s=1}^{n} |\ell(h(x_s), y) - \ell(h'(x_s), y_s) - (\ell(h(x_s), \hat{y}_s) - \ell(h'(x_s), \hat{y}_s))| + \ell(h(x_s), \hat{y}_s) + \ell(h'(x_s), \hat{y}_s)$$

$$= \tilde{A}_n(h, h') + \widehat{err}_n(h) + \widehat{err}_n(h').$$

Inserting this above and using the fact that $\mathbb{E}[g^+_{h,h'}] - \mathbb{E}[g^-_{h,h'}] = err(h) - err(h')$ finishes the proof. $\square$

Notice that the above bound has the noise term $A_n$ and its close cousin $\tilde{A}_n$. In the next section, we show how by using requesters functions, this noise term can be bounded, thereby resulting in a favorable generalization guarantee. For simplicity, throughout the rest of the paper, we will abbreviate $\Lambda_n(|H|^2)$ by $\Lambda_n$.

## A.2 Consistency and Controlling the Noise $A_n$.

We start by some necessacity definitions of slack terms used by the algorithm. The slack term used for trimming the version space $F_n$ is given by

$$\Delta'_n := \frac{2}{n}\left(\sqrt{\sum_{s=1}^{n} p_s} + 6\sqrt{\ln\left(\frac{(3+n)n^2}{\delta}\right)}\right) \times \ln\left(\frac{8n^2|F_1|^2\ln(n)}{\delta}\right)$$

and the slack term used in make the disagreement based requests is given by:

$$\tilde{\Delta}_n := \phi_n + \Lambda_n^2 + 2\Lambda_n\sqrt{\phi_n + \widehat{err}_n(h_+) + \widehat{err}_n(h_-)},$$

where $\phi_n$ is defined as follows:

$$\phi_n := \frac{8B^2}{n}\sum_{s=1}^{n}\Delta'_{s-1} + \frac{2B}{n}\ln\left(\frac{|F_1|^2 n(n+1)}{\delta}\right) + \frac{2\sqrt{B}}{n}\sqrt{\sum_{s=1}^{n}\Delta'_{s-1}}\sqrt{\ln\left(\frac{|F_1|^2 n(n+1)}{\delta}\right)}.$$

In the following lemma, we show a bound on the noise $A_n$ (and $\tilde{A}_n$) in terms of $\phi_n$. As we will see, the requester function $r_n$ are the key component in the proof of this lemma. In fact, the main role of the requester functions is to control the noise term $A_n$ so that it bounded by $\phi_n$. Notice that this directly implies that $\tilde{\Delta}_n$ is a upper bound on the right-hand side of the generalization guarantee of Corollary 1, excluding $err(h) - err(h')$, that can be empirically computed. At each round $n$, this empirically known generalization upper bound is leveraged by our algorithm to decide whether to request the label (i.e. disagreement based request) or infer the sign of $h^*(x_n)$ with high probability and use a pseudo-label that matches this sign. The lemma below will be used both in the subsequent consistency theorem and later also in the label complexity analysis.

**Lemma 2.** *Assume that for any $s \in [n]$, it holds that $(h^*, r^*) \in F_s$, $h^*$ is consistent on $\hat{S}_{s-1}$ and if a disagreement based request is not made for $y_s$, then $h^*$ is consistent on $\hat{S}_{s-1} \cup (x_s, \hat{y}_s)$. Then, for any $\delta > 0$, with probability at least $1 - \delta$,*

$$|A_n(h, h')| \leq \phi_n \ \text{ and } \ \tilde{A}_n(h, h') \leq \phi_n$$

*for any $h, h' \in H$.*

*Proof.* We start by deriving an intermediate bound on $A_n(h, h')$ for any $h, h' \in H$. First, we re-write the noise term as:

$$|A_n(h, h')| = \left| \frac{1}{n} \sum_{s=1}^n \ell(h(x_s), y_s) - \ell(h'(x_s), y_s) - (\ell(h(x_s), \hat{y}_s) - \ell(h'(x_s), \hat{y}_s)) \right|. \quad (2)$$

Each component in the sum of Equation (2) is non-zero only when $y_s \neq \hat{y}_s$. Note that $y_s \neq \hat{y}_s$ whenever the algorithm does not request, $r_s(x_s) \leq 0$, and whenever $y_s \neq \text{sgn}(h(x_s))$ for any $h$ consistent on $\hat{S}_s$.

Consider the pair $(h'_s, r_s) \in F_s$ where $h'_s$ is consistent on $\hat{S}_{s-1} \cup (x_s, \hat{y}_s)$. There always exists such a pair by the assumptions that $(h^*, r^*) \in F_s$ and $h^*$ is consistent on $\hat{S}_{s-1} \cup (x_s, \hat{y}_s)$. Using the fact that $\text{sgn}(h'_s(x_s)) = \hat{y}_s$ holds with probability $1 - \delta$, it then follows with probability $1 - \delta$ that

$$(2) = \left| \frac{1}{n} \sum_{s=1}^n (\ell(h(x_s), y_s) - \ell(h'(x_s), y_s) - (\ell(h(x_s), \hat{y}_s) - \ell(h'(x_s), \hat{y}_s))) \right.$$

$$\left. \mathbb{I}\{y_s \neq \hat{y}_s\} \mathbb{I}\{r_s(x_s) \leq 0\} \right|$$

$$\leq \left| \frac{1}{n} \sum_{s=1}^n 4B \mathbb{I}\{y_s \neq \text{sgn}(h'_s(x_s))\} \mathbb{I}\{r_s(x_s) \leq 0\} \right|, \quad (3)$$

where we bounded the loss differences by $4B$.

Next, we bound the expectation of the above random variable on the right-hand-side of (3) by using Lemma 6, which holds uniformly for all $(h, r) \in F_s$. Since $(h'_s, r_s) \in F_s$, Lemma 6 states that for any $\delta > 0$, with probability at least $1 - \delta$,

$$\mathbb{E}[\ell(y, h'_s(x)) \mathbb{I}\{r_s(x) \leq 0\}] \leq \min_{(h,r) \in F_s} \mathbb{E}[\ell(y, h(x)) \mathbb{I}\{r(x) \leq 0\}] + 2B\Delta'_{s-1}. \quad (4)$$

Using this fact and bounding the expectation of the right-hand-side of (3) by the surrogate loss $\ell$, it follows that

$$\mathbb{E}[\mathbb{I}\{y \neq \text{sgn}(h'_s(x)\} \mathbb{I}\{r_s(x) < 0\}] \leq \mathbb{E}[\ell(y, h'_s(x)) \mathbb{I}\{r_s(x) < 0\}]$$

$$\leq \min_{(h,r) \in F_s} \mathbb{E}[\ell(y, h(x)) \mathbb{I}\{r(x) \leq 0\}] + 2B\Delta'_{s-1}$$

$$= 2B\Delta'_{s-1}$$

where the last equality holds by Assumption 1.

The statement of the theorem on $A_n$ follows by using this inequality in conjunction with Inequality (3) and applying Lemma 3 of [Kakade and Tewari, 2009] to the martingale difference sequence

$$I_s = \mathbb{I}\{y_s \neq \text{sgn}(h'_s(x_s)\} \mathbb{I}\{r_s(x_s) < 0\} - \mathbb{E}\left[ \mathbb{I}\{y_s \neq \text{sgn}(h'_s(x_s)\} \mathbb{I}\{r_s(x_s) < 0\} | \mathcal{I}_{s-1} \right],$$

where $\mathcal{I}_s = \{(x_1, y_1, Q_1), \ldots, (x_s, y_s, Q_s)\}$ is the history up to time $s$. The above analysis can be similarly applied to $\tilde{A}_n$ to conclude the proof. $\square$

**Theorem 3.** *For any $\delta > 0$, with probability at least $1 - \delta$, it holds that for all $n > 0$:*

(1) $(h^*, r^*) \in F_n$;

(2) *$h^*$ is consistent on $\hat{S}_{n-1}$;*

(3) *if a disagreement based request is not made for $y_n$, $h^*$ is consistent on $\hat{S}_{n-1} \cup (x_n, \hat{y}_n)$. In particular, $h^*$ is consistent on $\hat{S}_n$.*

*Proof.* We proceed by strong induction over $n$. For the base case, consider $n = 1$. Here a disagreement-based request will always be made, so $h^*$ being consistent with $S_0 = \emptyset$ is clear. Also since $F_1 = H \times R$, it follows that $(h^*, r^*) \in F_1$. Assume now that the theorem holds for time steps $1, \ldots, n$.

To prove (1), by Lemma 5, it holds that for $(h', r') = \operatorname{argmin}_{(h,r) \in F_n} l_n(h, r)$,

$$l_n(h^*, r^*) - l_n(h', r') \leq \mathbb{E}[\ell_B(h^*(x), y)\mathbb{I}\{r^*(x) \leq 0\}] - \mathbb{E}[\ell_B(h'(x), y)\mathbb{I}\{r'(x) \leq 0\}] + \Delta'_n$$
$$\leq \Delta'_n.$$

The last inequality above follows since

$$\mathbb{E}[\ell_B(y, h^*(x))\mathbb{I}\{r^*(x) \leq 0\}] = \min_{(h,r) \in H \times R} \mathbb{E}[\ell_B(y, h(x))\mathbb{I}\{r(x) \leq 0\}]$$
$$\leq \mathbb{E}[\ell_B(y, h'(x))\mathbb{I}\{r'(x) \leq 0\}].$$

Thus, $(h^*, r^*) \in F_{n+1}$ by definition.

To prove (2) and (3), we need only consider the case where the algorithm is not querying due to disagreement, so without loss of generality, we assume that upon seeing $x_{n+1}$, we have

$$\widehat{err}_n(h_+) - \widehat{err}_n(h_-) > \tilde{\Delta}_n.$$

Our goal is then to show that $\operatorname{sgn} h^*(x_{n+1}) = -1$. For the sake of contradiction, assume the opposite. Then $\widehat{err}_n(h^*) \geq \widehat{err}_n(h_+)$. Using the above two conditions, it follows that

$$\widehat{err}_n(h^*) - \widehat{err}_n(h_-) = \widehat{err}_n(h^*) - \widehat{err}_n(h_+) + (\widehat{err}_n(h_+) - \widehat{err}_n(h_-))$$
$$> \widehat{err}_n(h^*) - \widehat{err}_n(h_+) + \tilde{\Delta}_n \geq \tilde{\Delta}_n$$
$$= \phi_n + \Lambda_n^2 + 2\Lambda_n\sqrt{\phi_n + \widehat{err}_n(h^*) + \widehat{err}_n(h_-)}.$$

At the same time, from our Corollary 1 and Lemma 2 which holds with probability at least $1 - \delta$ and by the inductive assumption, we have

$$err(h^*) - err(h_-) \geq \widehat{err}_n(h^*) - \widehat{err}_n(h_-) - \phi_n - \Lambda_n^2$$
$$- 2\Lambda_n\sqrt{\phi_n + \widehat{err}_n(h^*) + \widehat{err}_n(h_-)}.$$

Combined these two bounds would imply $err(h^*) > err(h_-)$, which is a contradiction. $\square$

### A.3 Final Generalization Bound

Given the above results, we are ready to prove the final generalization bound of the ALPS algorithm.

**Theorem 1.** *Under Assumption 1, for any $\delta > 0$, with probability at least $1 - \delta$, for any $n > 0$,*

$$err(h_n) \leq \nu + \tilde{O}\left(\sqrt{\frac{\nu}{n}} + \frac{1}{n}\right),$$

*where $h_n$ is the hypothesis computed by ALPS in round $n$.*

*Proof.* From Corollary 1, with probability at least $1 - \delta$,

$$err(h_n) \leq \nu + |A_n(h_n, h^*)| + \Lambda_n^2 + 2\Lambda_n\sqrt{\tilde{A}_n(h_n, h^*) + \widehat{err}_n(h_n) + \widehat{err}_n(h^*)}.$$

Focusing on the terms under the square root, since $\widehat{err}_n(h_n) \leq \widehat{err}_n(h^*)$ by definition, we need to bound $\widehat{err}_n(h^*)$. Since the main difference between $\widehat{err}_n(h^*)$ and $err_n(h^*)$ depends on how often the pseudo-labels differ from the true labels, it follows that $\widehat{err}_n(h^*) \leq err_n(h^*) + \frac{B}{n}\sum_{t=1}^n \mathbb{I}\{y_t \neq \hat{y}_t\}$. By Theorem 3, $h^*$ is consistent on the pseudo-labels, that is with probability $1 - \delta$, $\text{sgn}(h^*(x_t)) = \text{sgn}(\hat{y}_t)$ for any pseudo-label $\hat{y}_t$ for $t \in [n]$. Using this fact and since zero-one loss is bounded by the loss function $\ell$, it holds that $\frac{B}{n}\sum_{t=1}^n \mathbb{I}\{y_t \neq \hat{y}_t\} \leq B\,err_n(h^*)$. Putting the above together, $\widehat{err}_n(h^*) \leq (1 + B)\,err_n(h^*) \leq (1 + B)(\nu + \Lambda_n + \Lambda_n\sqrt{\nu})$, where the last inequality follows from using Lemma 1, which holds with probability $1 - \delta$. Thus,

$$err(h_n) \leq \nu + |A_n(h_n, h^*)| + 2\Lambda_n\sqrt{\tilde{A}_n(h_n, h^*)} + \tilde{O}(\Lambda_n^2 + \Lambda_n\sqrt{\nu})\,.$$

By using Theorem 3 and Lemma 2, we bound the $A_n$ and $\tilde{A}_n$ term as follows with probability at least $1 - \delta$,

$$err(h_n) \leq \nu + \phi_n + 2\Lambda_n\sqrt{\phi_n} + \tilde{O}(\Lambda_n^2 + \Lambda_n\sqrt{\nu})\,.$$

Recalling that $\phi_n = \frac{8B^2}{n}\sum_{s=1}^n \Delta'_{s-1} + \frac{2B}{n}\ln\left(\frac{|F_1|^2 n(n+1)}{\delta}\right) + \frac{2\sqrt{B}}{n}\sqrt{\sum_{s=1}^n \Delta'_{s-1}}\sqrt{\ln\left(\frac{|F_1|^2 n(n+1)}{\delta}\right)}$ and by the same reasoning as in Theorem 5 and by Assumption 1, it holds, with probability at least $1 - \delta$, that $\sum_{s'=1}^s \Delta'_{s'-1} = \tilde{O}(\log s)$. The statement then follows from $\Lambda_n = \tilde{O}(1/\sqrt{n})$, while absorbing constants factors of $\delta$ from the union bounds into log factors. $\qquad\square$

# B Proofs of the Label Complexity

In this appendix, we first bound the amount of labels requested due to disagreement-based requests and then bound the queries used for learning the requester functions.

## B.1 Bounding the disagreement-based requests

**Lemma 3.** *For any $\delta > 0$, with probability at least $1 - \delta$, letting $h^*(x_{n+1}) = \hat{y}$, the probability of requesting $y_{n+1}$ due to disagreement-based request is bounded as*

$$\mathbb{P}[\widehat{err}_n(h_{-\hat{y}}) - \widehat{err}_n(h_{\hat{y}}) \leq \tilde{\Delta}_n] \leq \mathbb{P}\left[err(h_{-\hat{y}}) = O\left(\nu + \Lambda_n^2\right)\right]\,.$$

*Proof.* We use several facts that hold concurrently with probability at least $1 - \delta$: The original generalization bound for a single hypothesis, Lemma 2, and Theorem 3.

Without loss of generality, we assume $h^*(x_{n+1}) = -1$ and that the label is requested. The hypothesis in the lemma statement is then $h_{-\hat{y}} = h_+$. We have

$$\widehat{err}_n(h_+) - \widehat{err}_n(h_-) \leq \phi_n + \Lambda_n^2 + 2\Lambda_n\sqrt{\phi_n + \widehat{err}_n(h_+) + \widehat{err}_n(h_-)}\,.$$

We lower bound the left-hand-side by $\widehat{err}_n(h_+) - \widehat{err}_n(h^*) = err_n(h_+) - err_n(h^*) - A_n(h_+, h^*) \geq err_n(h_+) - err_n(h^*) - \phi_n$ by Lemma 2 and Theorem 3. Then, we upper bound the right-hand-side by using $\widehat{err}_n(h_+) \leq (1 + B)\,err_n(h_+)$ and $\widehat{err}_n(h_-) \leq (1 + B)\,err_n(h^*)$ by a similar reasoning as in Theorem 1. Thus,

$$err_n(h_+) \leq err_n(h^*) + 2\phi_n + \Lambda_n^2 + 2\Lambda_n\sqrt{\phi_n + (1 + B)(err_n(h_+) + err_n(h^*))}\,.$$

Using $\sqrt{a + b} \leq \sqrt{a} + \sqrt{b}$ for positive $a, b$ and Lemma 7 (therein with $A = err_n(h_+)$), we then have

$$err_n(h_+) \leq err_n(h^*) + 2\phi_n + 4(2 + B)\Lambda_n^2 + 2\Lambda_n\sqrt{\phi_n + (1 + B)\,err_n(h^*)}$$
$$+ 2\Lambda_n\sqrt{(1 + B)(err_n(h^*) + 2\phi_n + \Lambda_n^2 + 2\Lambda_n\sqrt{\phi_n + (1 + B)\,err_n(h^*)})}\,.$$

We now use Lemma 1 on the true unknown empirical errors: $err_n(h_+) \geq err(h_+) - \Lambda_n\sqrt{err(h_+)}$ and $err_n(h^*) \leq \nu + \Lambda_n^2 + \Lambda_n\sqrt{\nu}$. We further repeatedly use the reduction $\Lambda_n^2\sqrt{\nu} \leq \nu + \Lambda_n^2$.

$$err(h_+) - \Lambda_n\sqrt{err(h_+)} = O\left(\nu + \Lambda_n^2 + \phi_n + \Lambda_n\sqrt{\nu} + \Lambda_n\sqrt{\phi_n + \Lambda_n\sqrt{\phi_n}}\right).$$

Putting the above together along with the definition of $\phi_n$ and using Lemma 7 once more finishes the proof. $\square$

**Definition 3.** *We define two metrics. The first on the space of hypotheses as*
$$\rho(h,h') := \mathbb{E}_{(x,y)\sim D}\left[|\ell(h(x),y) - \ell(h(x),y)|\right].$$

*The second metric is defined on the space $H \times R$:*
$$\rho'((h,r),(h',r')) = \mathbb{E}[|\ell_B(h(x),y)\mathbb{I}\{r(x) \leq 0\} - \ell_B(h'(x),y)\mathbb{I}\{r'(x) \leq 0\}|].$$

*Using each metric, we define the $\gamma$-ball around the best-in-class as*
$B_\gamma(h^*) = \{h \in H : \rho(h,h^*) \leq \gamma\}$ *and* $B'_\gamma(h^*,r^*) = \{(h,r) \in H \times R : \rho'((h,r),(h^*,r^*)) \leq \gamma\}$ .

With these metrics in place, we introduce the following disagreement coefficients. These can be thought of as two different generalizations of the binary disagreement used for the zero-one loss, each tailored to one of the two notions of consistency or disagreement used in the analysis.

**Definition 4.** *Let the disagreement coefficient $\theta$ with respect to $H$ be*
$$\theta := \sup_{\gamma > 0}\left\{\frac{\mathbb{P}_x[\exists h \in B_r(h^*) : \mathrm{sgn}(h(x)) \neq \mathrm{sgn}(h^*(x))]}{\gamma}\right\}.$$

*Let the disagreement coefficient $\theta'$ with respect to $H \times R$ be*
$$\theta' = \inf_{\theta''}\left\{\forall \gamma \geq 0,\right.$$
$$\left.\mathbb{E}_x\left[\sup_{(h,r) \in B_\gamma(h^*,r^*)}\sup_y |\ell(h(x),y)\mathbb{I}\{r(x) \leq 0\} - \ell(h^*(x),y)\mathbb{I}\{r^*(x) \leq 0\}|\right] \leq \theta''\gamma\right\},$$

**Lemma 4.** *With probability $1 - \delta$ it holds for every $n$, that letting $h^*(x_{n+1}) = \hat{y}$, we have*
$$\mathbb{P}[err(h_{-\hat{y}}) \leq \eta] \leq \theta(\nu + \eta)$$
*for any $\eta > 0$.*

*Proof.* Consider the high probability event that $h^*$ is consistent with $\hat{S}_n$, which holds due to Theorem 3, and consider the case where a request is made. From the definition of $\rho$,
$$\rho(h_{-\hat{y}}, h^*) \leq \nu + err(h_{-\hat{y}}).$$
By this inequality, we get
$$\begin{aligned}\mathbb{P}[err(h_{-\hat{y}}) \leq \eta] &\leq \mathbb{P}[\rho(h_{-\hat{y}}, h^*) \leq \nu + \eta]\\&\leq \mathbb{P}[\exists h \in B_{\nu+\eta}(h^*) : \mathrm{sgn}(h(x)) \neq \mathrm{sgn}(h^*(x))]\\&\leq \theta(\nu + \eta),\end{aligned}$$
where the second inequality uses that $h_{-\hat{y}}$ and $h^*$ disagree on $x_{n+1}$ by construction and the final inequality uses the disagreement coefficient of Definition 4. $\square$

**Theorem 4.** *For any $\delta > 0$, with probability at least $1 - \delta$, for all $n > 0$, the expected number of queried labels due to disagreement-based request is bounded by*
$$\mathbb{E}\left[\sum_{s=1}^n \mathbb{P}[\widehat{err}_{s-1}(h_{-\hat{y}}) - \widehat{err}_{s-1}(h_{\hat{y}}) \leq \tilde{\Delta}_{s-1}]\right] = O\left(\theta\nu n + \theta\log^2 n\right),$$
*where the expectation is with respect to the draws of all $(x_s, y_s)$.*

*Proof.* By combining Lemmas 3 and Lemma 4, with probability at least $1 - \delta$ for all $n$,
$$\mathbb{P}[\widehat{err}_{n-1}(h_{-\hat{y}}) - \widehat{err}_{n-1}(h_{\hat{y}}) > \tilde{\Delta}_{n-1}] = O\left(\theta(\nu + \Lambda_n^2)\right).$$
Summing these terms yields the above rates as $\Lambda_n^2 = O(\ln(n)/n)$. $\square$

## B.2 Bounding the component related to learning requester functions

To learn the requester functions, we use importance weighted estimates and so we seek a bound on $\sum_{s=1}^{n} Q_s$, where we recall each $Q_s$ is coin flip with probability $p_s$.

**Theorem 5.** *For all $\delta > 0$, for all $n \geq 0$, with probably at least $1 - \delta$, it holds that*

$$\sum_{s=1}^{n} Q_s \leq 8\theta' \left( \mathbb{E}[\ell(y, h^*(x))\mathbb{I}\{r^*(x) \leq 0\}]n + O(B\sqrt{\mathbb{E}[\ell(y, h^*(x))\mathbb{I}\{r^*(x) \leq 0\}]n \log(n|F_1|/\delta)}) \right)$$
$$+ O(B\log^3(n|F_1|/\delta)).$$

*Proof.* First, we prove a bound on $\mathbb{E}[p_s]$. By definition of $\rho'$, it holds that $\rho'((h, r), (h^*, r^*)) \leq \mathbb{E}[\ell_B(y, h(x))\mathbb{I}\{r(x) \leq 0\}] + \mathbb{E}[\ell_B(y, h^*(x))\mathbb{I}\{r^*(x) \leq 0\}]$. For any pair $(h, r) \in F_s$, $\mathbb{E}[\ell_B(y, h(x))\mathbb{I}\{r(x) \leq 0\}] \leq \mathbb{E}[\ell_B(y, h^*(x))\mathbb{I}\{r^*(x) \leq 0\}] + 2\Delta'_{s-1}$ by Lemma 6. Hence, $F_s \subseteq B_\gamma(h^*, r^*)$ where $\gamma = 2\mathbb{E}[\ell_B(y, h^*(x))\mathbb{I}\{r^*(x) \leq 0\}] + 2\Delta'_{s-1}$. Then by definition of disagreement coefficient:

$$\mathbb{E}[p_s] \leq \mathbb{E}[\max_{(h,r),(h',r') \in F_s} \max_y \ell_B(y, h(x_s))\mathbb{I}\{r(x_s) \leq 0\} - \ell_B(y, h'(x_s))\mathbb{I}\{r'(x_s) \leq 0\}]$$

$$\leq 2\mathbb{E}[\max_{(h,r) \in F_s} \max_y \ell_B(y, h(x_s))\mathbb{I}\{r(x_s) \leq 0\} - \ell_B(y, h^*(x_s))\mathbb{I}\{r^*(x_s) \leq 0\}]$$

$$\leq 2\mathbb{E}[\max_{(h,r) \in B_\gamma(h^*, r^*)} \max_y \ell_B(y, h(x_s))\mathbb{I}\{r(x_s) \leq 0\} - \ell_B(y, h^*(x_s))\mathbb{I}\{r^*(x_s) \leq 0\}]$$

$$\leq 2\theta'\gamma.$$

Converting from $\ell_B$ to the surrogate loss $\ell$ and using the definition of $\gamma$, we find that $\mathbb{E}[p_s] \leq 4\theta'(\mathbb{E}[\ell(y, h^*(x))\mathbb{I}\{r^*(x) \leq 0\}] + B\Delta'_{s-1})$.

Using the above and by straightforward modifications of Lemma 6 in [Cortes et al., 2019b] and Theorem 1 in [Cortes et al., 2019b] concludes the proof. $\qquad\square$

## B.3 Final Label Complexity

Putting the above together, we attain the final label complexity guarantee.

**Theorem 2.** *Under Assumption 1, for any $\delta > 0$, with probability $1 - \delta$, the label complexity of the ALPS algorithm at time $n > 0$ is bounded as $\tilde{O}(n\nu(\theta + C))$.*

*Proof.* The label complexity component is the addition of Theorem 5, Theorem 4 and the term $\mathbb{E}[\mathbb{I}\{r_n(x) > 0\}]$. Since $(h^*, r^*) \in F_n$ for all $n$ with high probability by Theorem 3, then it must be the case that $r^* \in R_n$. This in turn implies that $\mathbb{E}[\mathbb{I}\{r_n(x) > 0\}] \leq \mathbb{E}[\mathbb{I}\{r^*(x) > 0\}]$ since $r_n = \operatorname{argmin}_{r \in R_n} \mathbb{E}[\mathbb{I}\{r(x) > 0\}]$. $\qquad\square$

# C  Some technical lemmas

We first present two technical lemma for the importance weighted estimates $l_{n-1}(h, r)$ of $\mathbb{E}[\ell_B(y, h(x))\mathbb{I}\{r(x) \leq 0\}]$ in term of the slack term $\Delta'_n$.

**Lemma 5.** *For any $\delta > 0$, with probability at least $1 - 2\delta$, for all $n \geq 3$, for all $(h, r), (h', r') \in F_n$,*

$$|l_n(h, r) - l_n(h', r') - \mathbb{E}[\ell_B(y, h(x))\mathbb{I}\{r(x) \leq 0\}] + \mathbb{E}[\ell_B(y, h'(x))\mathbb{I}\{r'(x) \leq 0\}]| \leq \Delta'_n.$$

*Proof.* For $s \in [n]$, consider the random variables:

$$Z_s = \frac{Q_s}{p_s}(\ell_B(y_s, h(x_s))\mathbb{I}\{r(x_s) \leq 0\} - \ell_B(y_s, h'(x_s))\mathbb{I}\{r'(x_s) \leq 0\})$$
$$- \mathbb{E}[\ell_B(y, h(x))\mathbb{I}\{r(x) \leq 0\}] + \mathbb{E}[\ell_B(y, h'(x))\mathbb{I}\{r'(x) \leq 0\}],$$

for any $(h, r), (h', r') \in F_n$.

Let $\mathcal{I}_n = \{(x_1, y_1, Q_1), \ldots, (x_n, y_n, Q_n)\}$ be the history up to time $n$. By applying Lemma 4 in [Cortes et al., 2019b], for any $0 < \delta < 1$, and $n \geq 3$, with probability at least $1 - \delta$,

$$\left| \sum_{s=1}^{n} Z_s \right| \leq \max \left\{ 2\sqrt{\sum_{s=1}^{n} \mathbb{E}_{x_s}[p_s | \mathcal{I}_{s-1}]}, 6\sqrt{\log\left(\frac{8\log(n)}{\delta}\right)} \right\}$$
$$\times \sqrt{\log\left(\frac{8\log(n)}{\delta}\right)}.$$

Taking a union bound over $n \geq 3$ and all $(h, r), (h', r') \in F_1$, it holds that

$$|l_n(h, r) - l_n(h', r') - \mathbb{E}[\ell_B(y, h(x))\mathbb{I}\{r(x) \leq 0\}] + \mathbb{E}[\ell_B(y, h'(x))\mathbb{I}\{r'(x) \leq 0\}]|$$

$$\leq \frac{1}{n} \max \left\{ 2\sqrt{\sum_{s=1}^{n} \mathbb{E}_{x_s}[p_s | \mathcal{I}_{s-1}]}, 6\sqrt{\log\left(\frac{8n^2|F_1|^2 \log(n)}{\delta}\right)} \right\}$$
$$\times \sqrt{\log\left(\frac{8n^2|F_1|^2 \log(n)}{\delta}\right)}.$$

Via Proposition 2 in [Cesa-Bianchi and Gentile, 2008], with probability at least $1 - \delta$, for all $n \geq 3$,

$$\sum_{s=1}^{n} \mathbb{E}_{x_s}\left[p_s | \mathcal{I}_{t-1}\right] \leq \left(\sum_{s=1}^{n} p_s\right) + 36\log\left(\frac{(3 + \sum_{s=1}^{n} p_s)n^2}{\delta}\right)$$

$$+ 2\sqrt{\left(\sum_{s=1}^{n} p_s\right)\log\left(\frac{(3 + \sum_{s=1}^{n} p_s)n^2)}{\delta}\right)}$$

$$\leq \left(\sqrt{\sum_{s=1}^{n} p_s} + 6\sqrt{\log\left(\frac{(3 + n)n^2}{\delta}\right)}\right)^2.$$

Combining the above, we attain the desired bound. $\qquad\square$

**Lemma 6.** *For any $\delta > 0$, with probability $1 - \delta$, for any $s > 0$ and any $(h, r), (h', r') \in F_s$,*

$$\mathbb{E}[\ell(y, h(x))\mathbb{I}\{r(x) \leq 0\}] \leq \mathbb{E}[\ell(y, h'(x))\mathbb{I}\{r'(x) \leq 0\}] + 2B\Delta'_{s-1}.$$

*In particular, it holds that*

$$\mathbb{E}[\ell(y, h(x))\mathbb{I}\{r(x) \leq 0\}] \leq \min_{(h,r) \in F_s} \mathbb{E}[\ell(y, h(x))\mathbb{I}\{r(x) \leq 0\}] + 2B\Delta'_{s-1}.$$

*Proof.* Take any $(h, r), (h', r') \in F_n$, then by using Lemma 5 and since $F_n \subseteq F_{n-1}$,

$$\mathbb{E}[\ell_B(y, h(x))\mathbb{I}\{r(x) \leq 0\}] - \mathbb{E}[\ell_B(y, h'(x))\mathbb{I}\{r'(x) \leq 0\}]$$
$$\leq l_{s-1}(h, r) - l_{s-1}(h', r') + \Delta'_{s-1}$$
$$\leq \min_{(h,r) \in F_{s-1}} l_{s-1}(h, r) + \Delta'_{s-1} - \min_{(h,r) \in F_{s-1}} l_{s-1}(h, r) + \Delta'_{s-1}$$
$$\leq 2\Delta'_{s-1},$$

where we used the fact that $l_{s-1}(h, r) \leq \min_{(h,r) \in F_{s-1}} l_{s-1}(h, r) + \Delta'_{s-1}$. By multiplying by $B$, the bound of the lemma directly follows. $\qquad\square$

Next consider a technical lemma used throughout the analysis, which will allow us to exchange terms in the fast rate bounds by adding $1/n$ (corresponding to $C^2$).

**Lemma 7.** *For non-negative real numbers $A, B, C \geq 0$ we have*

$$A \leq B + C\sqrt{A} \quad \Rightarrow \quad A \leq B + C^2 + C\sqrt{B}.$$

*Proof.* Since it holds that $A \leq B + C\sqrt{A}$, it follows that $\sqrt{A}$ is less than the largest root of the equation $x^2 - B - Cx = 0$, that is

$$\sqrt{A} \leq \frac{C + \sqrt{C^2 + 4B}}{2}.$$

The second inequality in the lemma follows from squaring this and using $\sqrt{y + y'} \leq \sqrt{y} + \sqrt{y'}$ for positive $y, y'$. $\qquad\square$

| Dataset | # features | # unlabeled examples | # test examples | Notes |
|---|---|---|---|---|
| ijcnn | 22 | 49990 | 91701 | |
| satimage | 36 | 4435 | 2000 | 'red soil' vs. rest |
| cod-rna | 8 | 59535 | 271617 | |
| mnist | 780 | 60000 | 10000 | 'odd' vs. 'even' |
| cifar | 3072 | 12000 | 2000 | 'horse' vs. 'ship' |
| acoustic | 50 | 78823 | 19705 | '1' vs. rest |
| german | 24 | 500 | 500 | |
| pima | 8 | 500 | 228 | |
| gestures | 33 | 5000 | 4873 | 'rest position' vs. rest |
| phishing | 68 | 5000 | 6055 | |
| shuttle | 9 | 43500 | 14500 | 'rad flow' vs. rest |
| skin | 3 | 100,000 | 145057 | |
| australian | 14 | 500 | 190 | |
| breastcancer | 10 | 500 | 183 | |
| guide | 4 | 3089 | 4000 | |
| a9a | 123 | 32561 | 16281 | |
| mushroom | 112 | 5119 | 3005 | |
| splice | 60 | 2000 | 1175 | |

Table 1: Dataset statistics including number of features, number of unlabeled examples, number of test examples and notes on the classification task.

## D   Experimental Supplements

We plot the performance of the ALPS algorithm, the IWAL algorithm, the ex-post best margin algorithm, and the passive learning algorithm in Figure 3,4, and 5 for 3-layer neural networks and in Figure 9,10, and 11 for 4-layer neural networks. Overall, ALPS outperforms all baselines on almost all datasets for both types of networks. In a few cases, we find that the ex-post best margin outperforms ALPS with respect to the accuracy curve, but not with respect to logistic loss (e.g., german). We also find that IWAL does not attain the accuracy or logistic loss of the passive learning curve on a small subset of the datasets.

For each dataset, first each feature was normalized to have zero mean and unit variance, and then all examples in the dataset were divided by the maximum $l_2$-norm of the examples. That is, letting $D$ be the dataset, each data point in the dataset, $x_i \in D$, was divided by $\max_{i \in D} \|x_i\|_2$. Table 1 contains the main statistics of each dataset. The datasets were taken from the LIBSVM dataset library, the OpenML dataset library, and MNIST and CIFAR databases Vanschoren et al. [2013], Chang and Lin [2011], Krizhevsky [2009], LeCun and Cortes [2010]. If available, the train/test split in these libraries was used. Most of the datasets admit binary labels, but there is a subset of multi-class datasets, which were turned into binary classification tasks. Specifically, for gestures, satimage, acoustic, and shuttle, the binary classification task was separating one-versus-rest for a given fixed class (see table for which class). For mnist, the binary task was separating odd versus even digits and for cifar, the binary task was separating 'horse' and 'ship' images.

To generate the hypothesis functions for each type of neural network models, we trained the respective model on random subsets of the training data while varying hyperparamters. The points used in the random training subsets were then removed from the unlabeled pool. The Multi-layer Perceptron algorithm in scikit-learn library was used with solver=lbfgs and maximum number of iterations set to 1,000 Pedregosa et al. [2011] . The number of hidden units in the 3-layer neural network was $(50, 10, 5)$, while in the 4-layer network it was $(10, 10, 10, 10)$. For each dataset, we generated two hypothesis set, one for 3-layer networks and one for 4-layer networks, and tested the algorithms on both hypothesis sets independently to see the effects the model has on the active learning algorithms. In order to create hypothesis set $H$, we generated 1,000 hypothesis functions by training the the Multi-layer Perceptron algorithm on random subsets of between 50 and 500 points while randomly choosing an $l_2$-norm regularization parameter (alpha) from the set $\{2^i\}_{i=-4}^4$ and randomly initializing with an integer in $\{1, \cdots, 100\}$ the random number generation (random_state) for weights and bias initialization. For the smaller datasets (australian, german, breastcancer and pima), the random training subsets used for training were instead taken between 50 and 100 points.

For the ALPS algorithm, the set of thresholds used was $\Gamma_r = \{0.1, 0.25, 0.5, 0.75, 0.9\}$. For the margin algorithm, we defined the threshold set $\Gamma_m$ via the following tuning procedure. We started by running margin algorithms with $\Gamma_m = \Gamma_r$, which resulted in algorithms that requested too few points admitting poor accuracy. Thus, the threshold values were enlarged until good accuracy learning curves were reached for nine threshold values. The tuned threshold set turned out to be $\Gamma_m = \{\sum_{i=1}^{j} 9(1/10)^i\}_{j=1}^9$. We found that the neural networks were overly confident in their predictions resulting in high threshold values. Notice that the above tuning clearly gives an unfair advantage to the margin algorithms over the other algorithms, but it gives us an upper bound on the best possible performance in this setting.

The best exp-post margin algorithm is chosen as as follows. For all margin algorithms with thresholds in $\Gamma_m$ and for the passive learning algorithm, we calculated the area under the learning curve of accuracy versus the number of observed points by using the Trapezoid Rule. The best exp-post margin algorithm is the one with smallest threshold that admits an learning curve area that is within one standard error of the passive learning curve area. If no algorithm exists, the best exp-post margin algorithm is the one with the largest learning curve area.

We report the result of the margin algorithm for all these threshold values in Figure 6,7, and 8 for 3-layer networks and in Figure 12,13, and 14 for 4-layer networks. For the legend, the margin algorithms are enumerated in order from the smallest to largest threshold in set $\Gamma_m$. That is, Margin-1 corresponds to the smallest threshold, Margin-2 to the second smallest, etc. The generalization ability of the margin algorithm largely varies with the threshold values. In almost all datasets, there exist thresholds whose corresponding margin algorithm admits an accuracy curve that is below that of passive learning and also thresholds that are on par with passive learning, thereby indicating that we tested a good set of threshold values.

For experiments with larger neural networks, ALPS can be run in the online batch setting of Amin et al. [2020] where the streaming algorithm's internal state is frozen until a batch of points is selected. This would considerably dampen the larger networks' longer training and inference times.

Note that we do not compare to the region-based algorithms of Cortes et al. [2019b, 2020] since these algorithms select hypothesis functions outside the class $H$. Nonetheless, both algorithms in Cortes et al. [2019b, 2020] are meta-algorithms that use IWAL as a subroutine and one can directly derive learning guarantees for region-based algorithms that instead use ALPS as a subroutine.

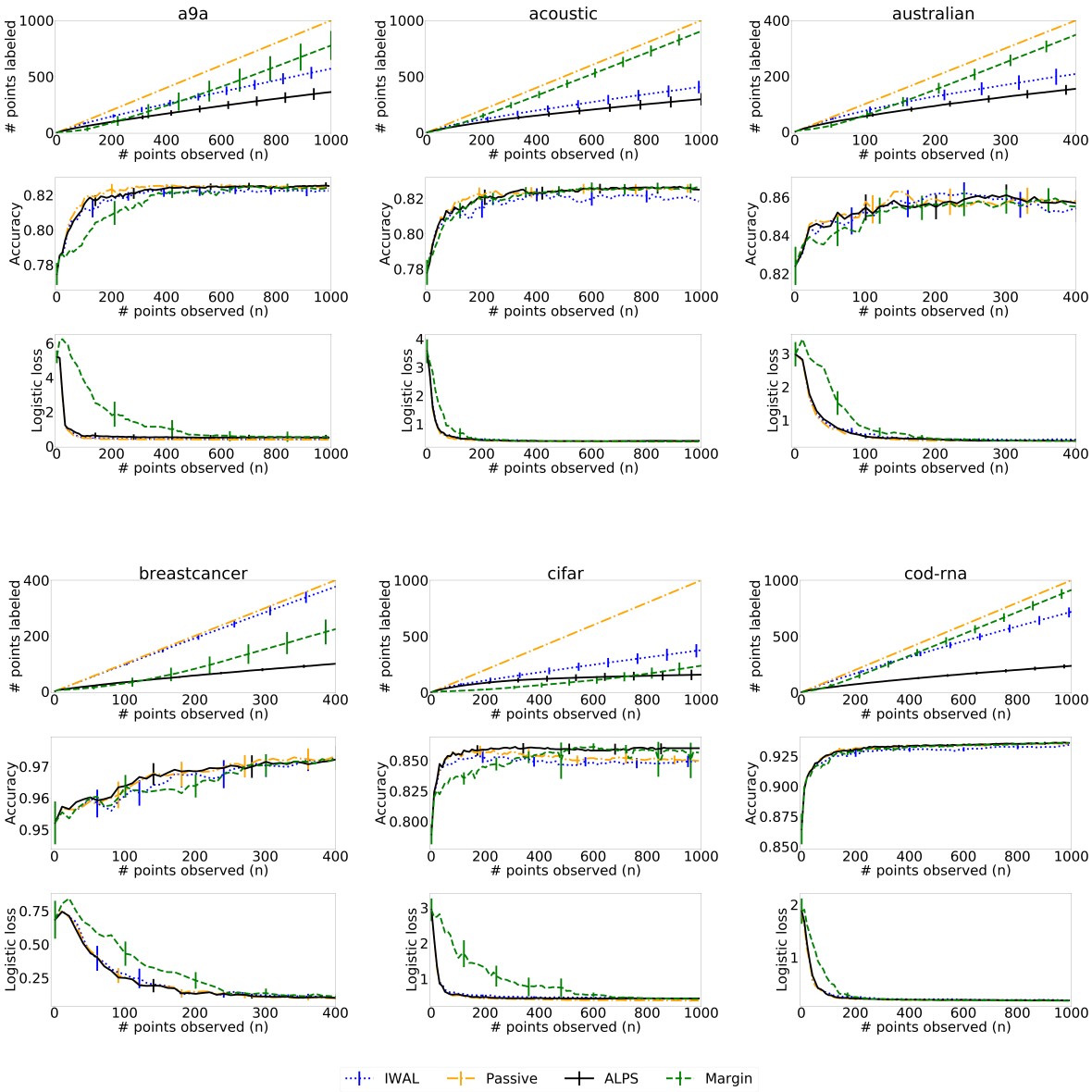

Figure 3: Comparison of the ALPS algorithm, the IWAL algorithm, the ex-post best margin algorithm and the passive learning algorithm for 3-layer networks (1/3). The figures shows the mean over the 50 trials and twice its standard error of the number of labeled points, accuracy and logistic loss on the test set of the model returned by each algorithm as a function of the number of observed points (or rounds) $n$.

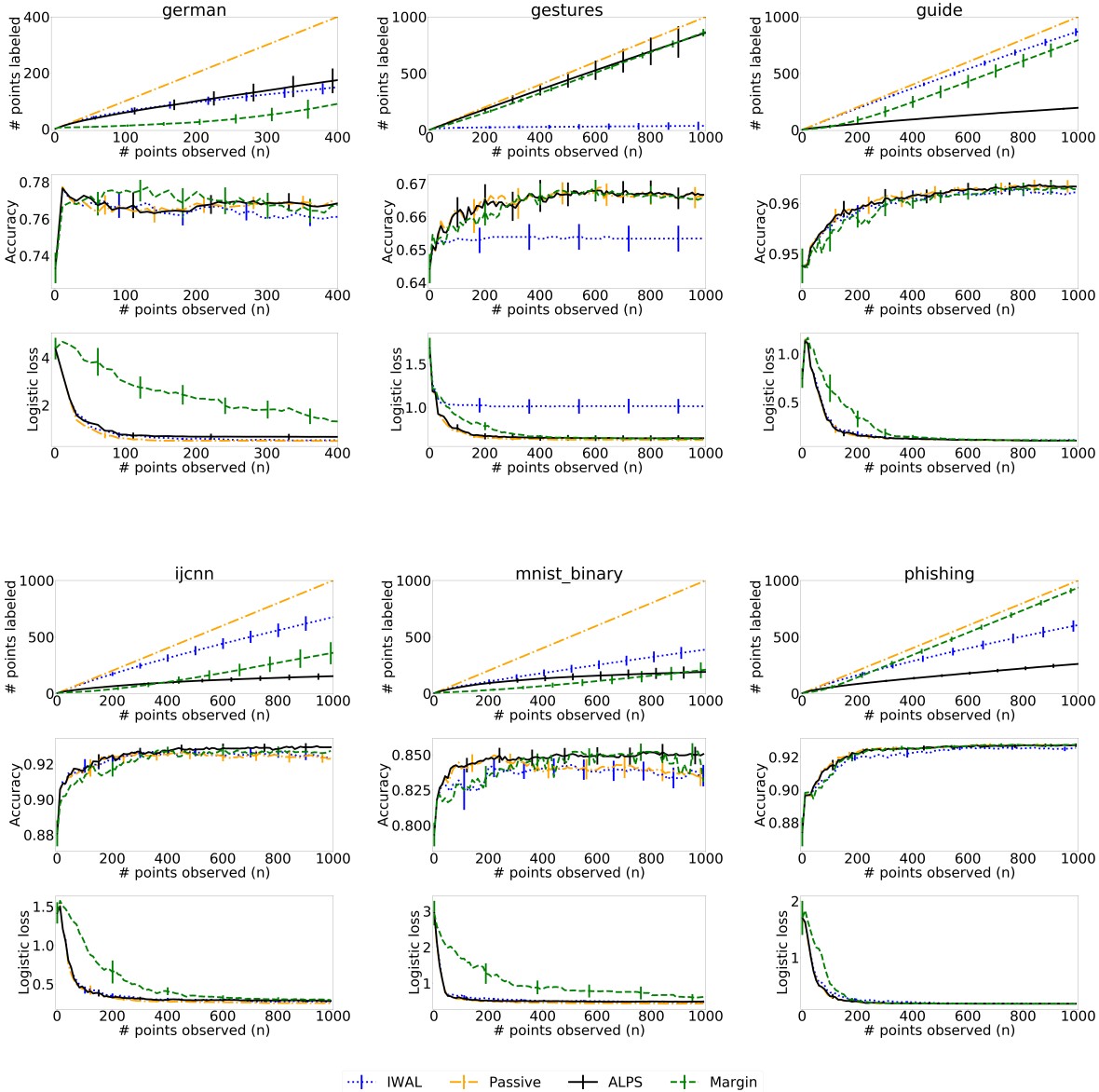

Figure 4: Comparison of the ALPS algorithm, the IWAL algorithm, the ex-post best margin algorithm and the passive learning algorithm for 3-layer networks (2/3). The figures shows the mean over the 50 trials and twice its standard error of the number of labeled points, accuracy and logistic loss on the test set of the model returned by each algorithm as a function of the number of observed points (or rounds) $n$.

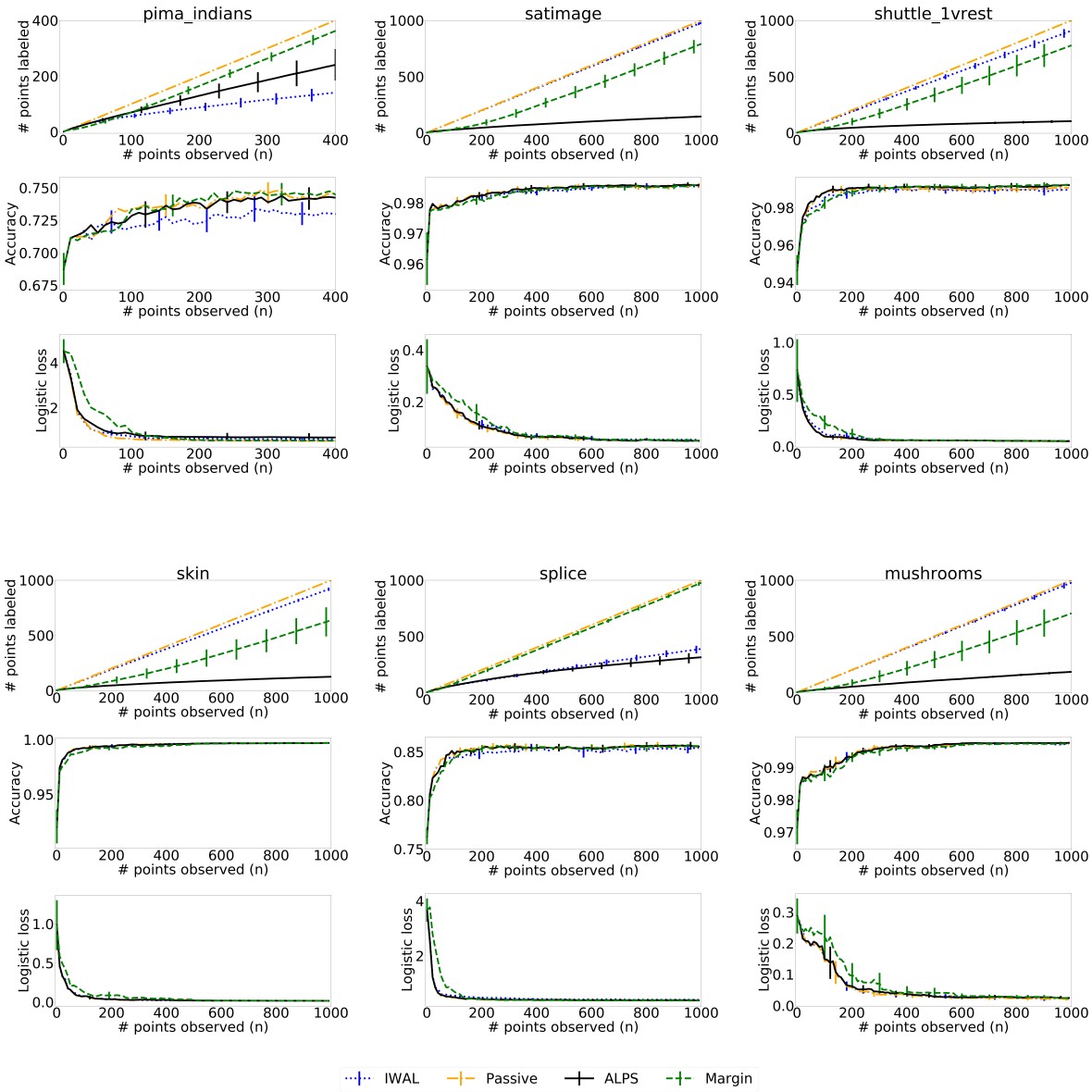

Figure 5: Comparison of the ALPS algorithm, the IWAL algorithm, the ex-post best margin algorithm and the passive learning algorithm for 3-layer networks (3/3). The figures shows the mean over the 50 trials and twice its standard error of the number of labeled points, accuracy and logistic loss on the test set of the model returned by each algorithm as a function of the number of observed points (or rounds) $n$.

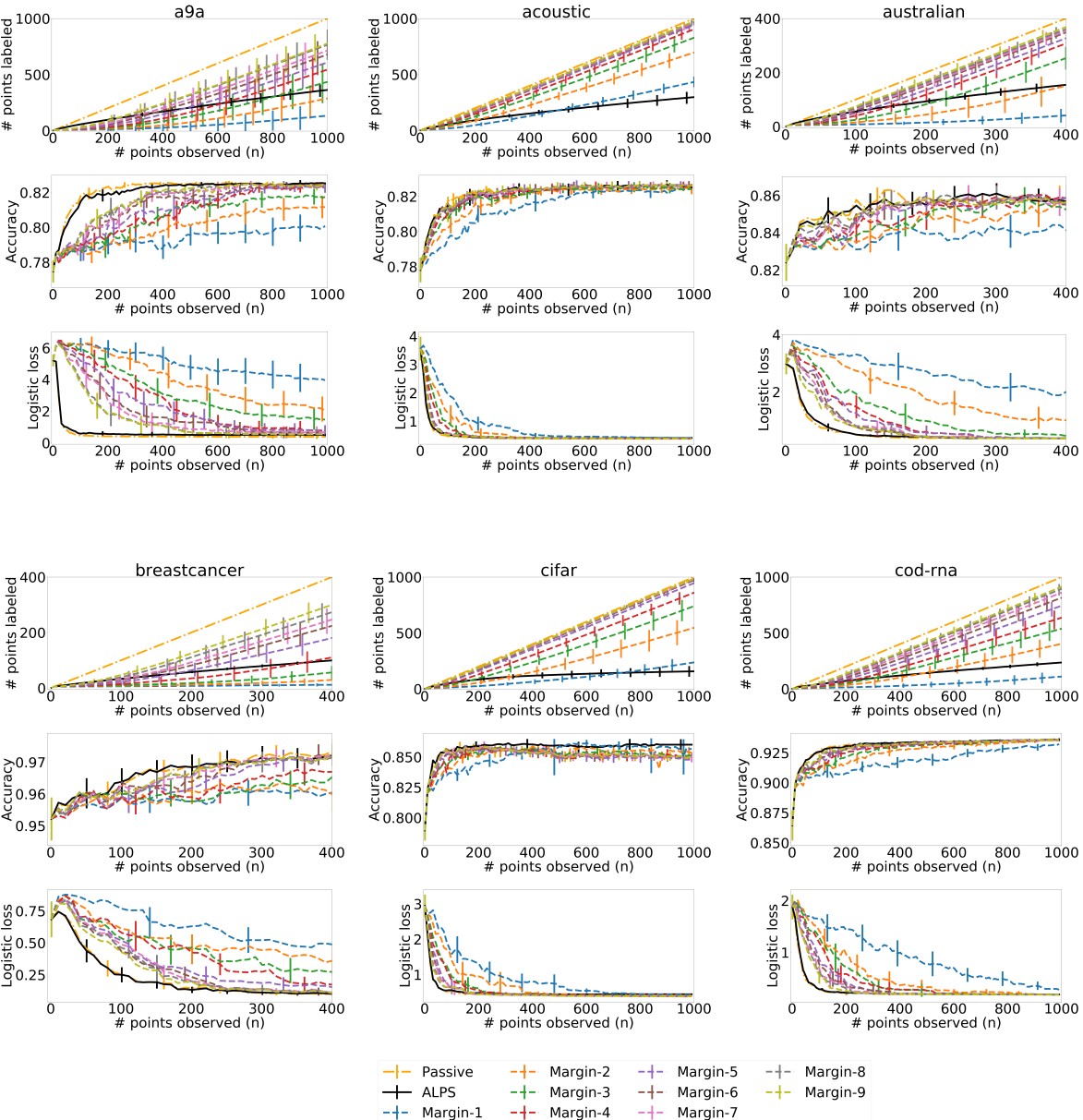

Figure 6: Comparison of all nine margin algorithms to passive learning and ALPS for 3-layer networks (1/3). The margin algorithms are ordered from the smallest to the largest threshold in $\bar{\Gamma}_m$, where Margin-1 corresponds to the smallest threshold, Margin-2 to the second smallest, etc. The figures shows the mean over the 50 trials and twice its standard error of the number of labeled points, accuracy and logistic loss on the test set of the model returned by each algorithm as a function of the number of observed points (or rounds) $n$.

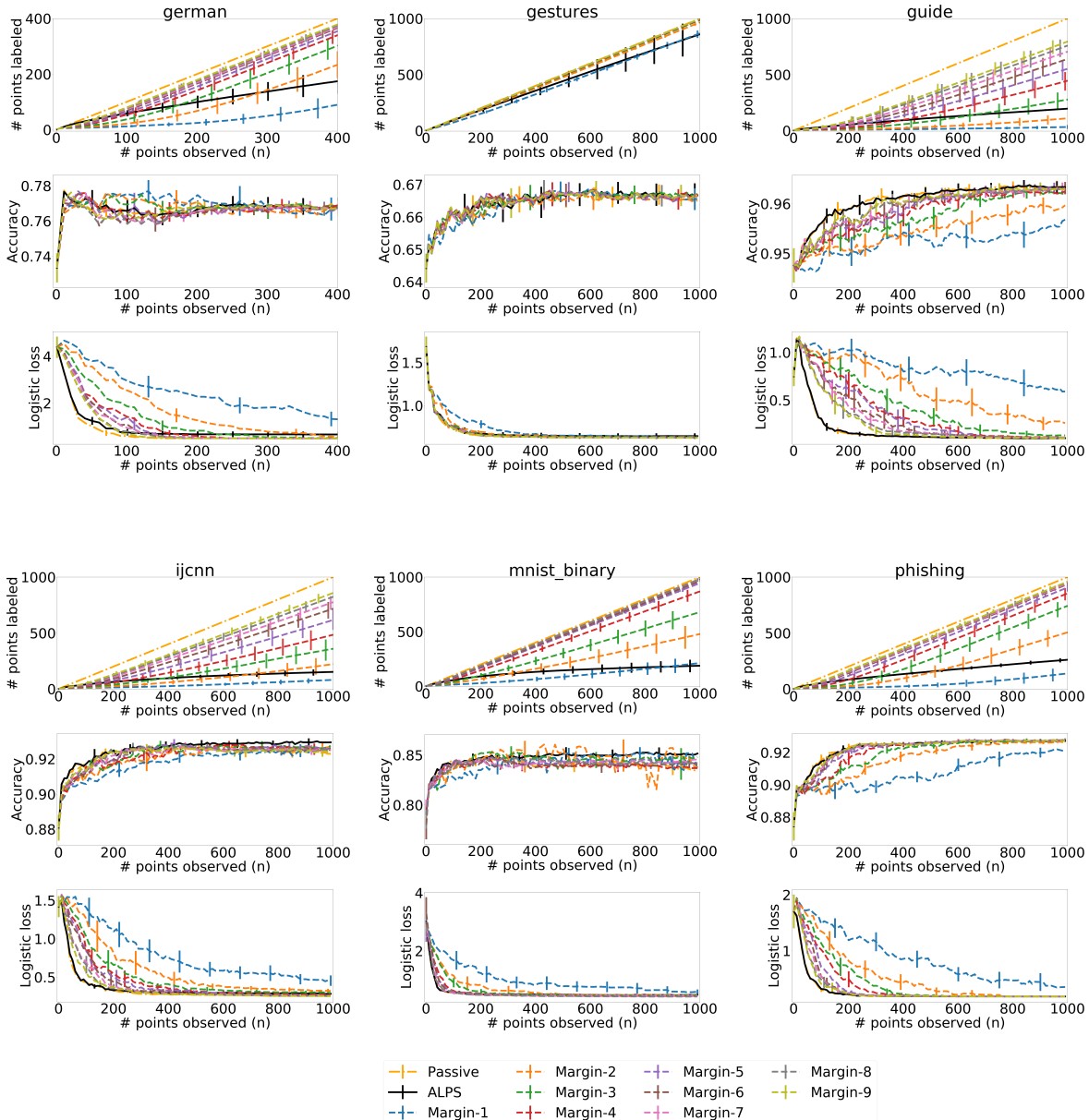

Figure 7: Comparison of all nine margin algorithms to passive learning and ALPS for 3-layer networks (2/3). The margin algorithms are ordered from the smallest to the largest threshold in $\bar{\Gamma}_m$, where Margin-1 corresponds to the smallest threshold, Margin-2 to the second smallest, etc. The figures shows the mean over the 50 trials and twice its standard error of the number of labeled points, accuracy and logistic loss on the test set of the model returned by each algorithm as a function of the number of observed points (or rounds) $n$.

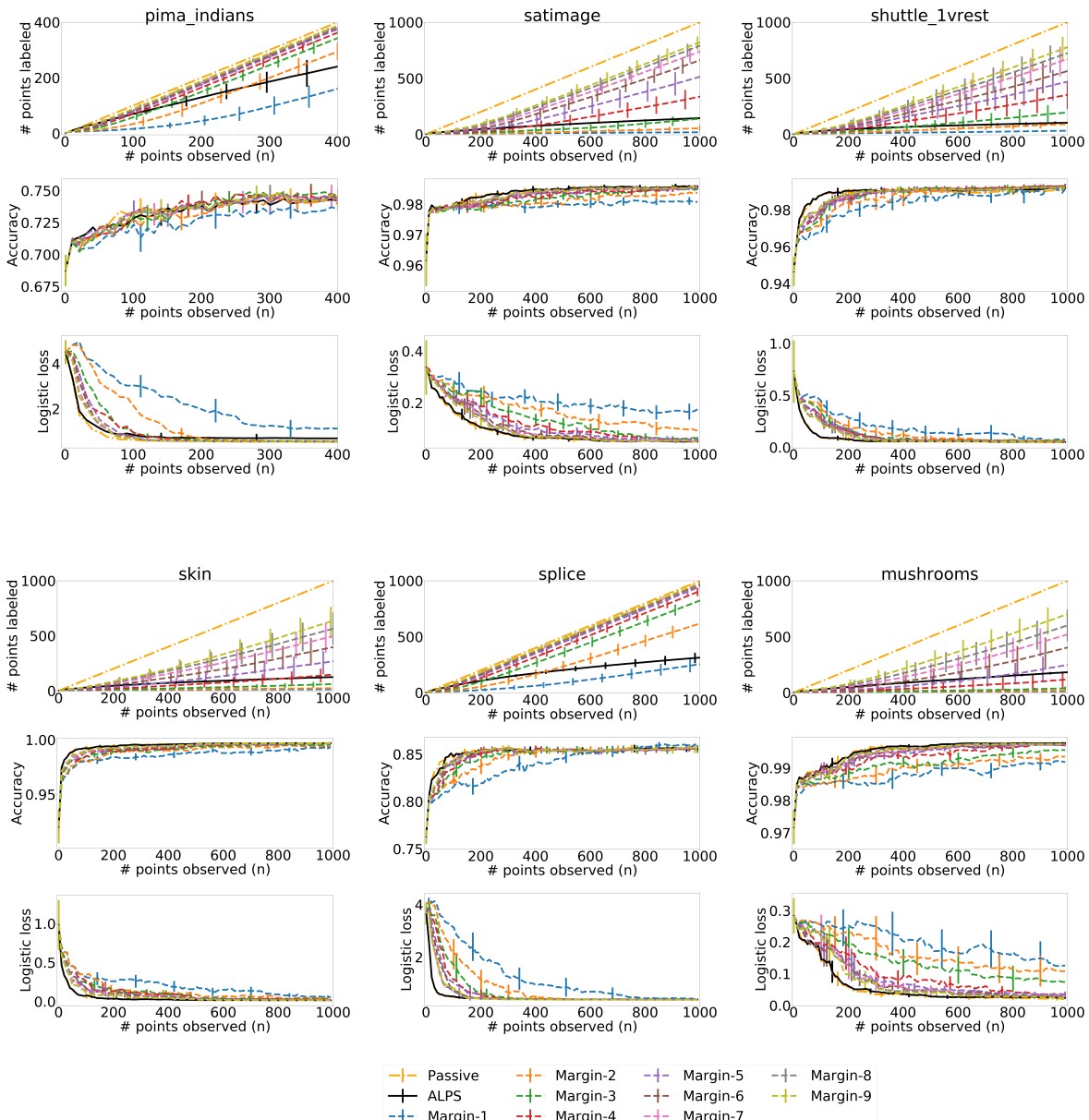

Figure 8: Comparison of all nine margin algorithms to passive learning and ALPS for 3-layer networks (3/3). The margin algorithms are ordered from the smallest to the largest threshold in $\bar{\Gamma}_m$, where Margin-1 corresponds to the smallest threshold, Margin-2 to the second smallest, etc. The figures shows the mean over the 50 trials and twice its standard error of the number of labeled points, accuracy and logistic loss on the test set of the model returned by each algorithm as a function of the number of observed points (or rounds) $n$.

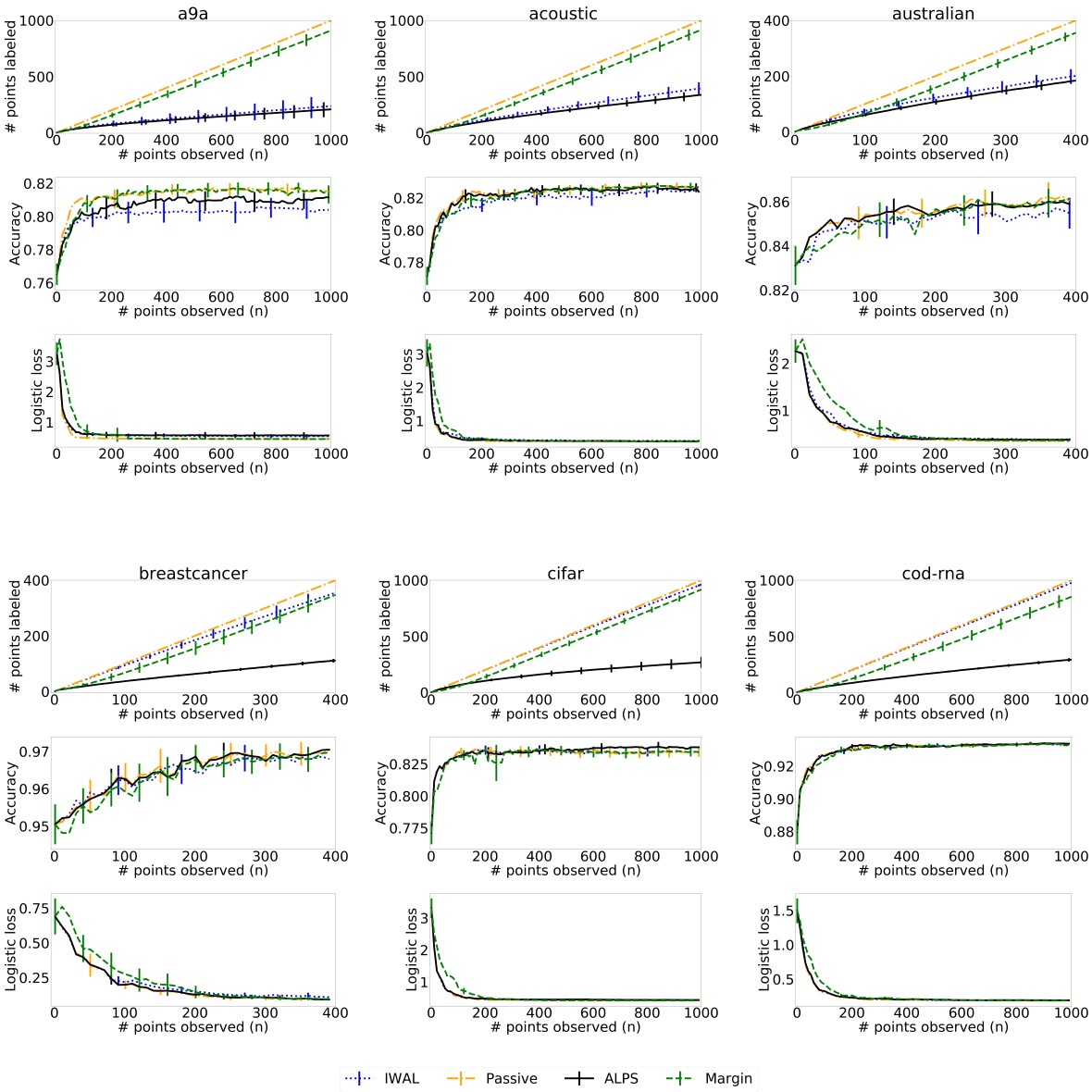

Figure 9: Comparison of the ALPS algorithm, the IWAL algorithm, the ex-post best margin algorithm and the passive learning algorithm for 4-layer networks (1/3). The figures shows the mean over the 50 trials and twice its standard error of the number of labeled points, accuracy and logistic loss on the test set of the model returned by each algorithm as a function of the number of observed points (or rounds) $n$.

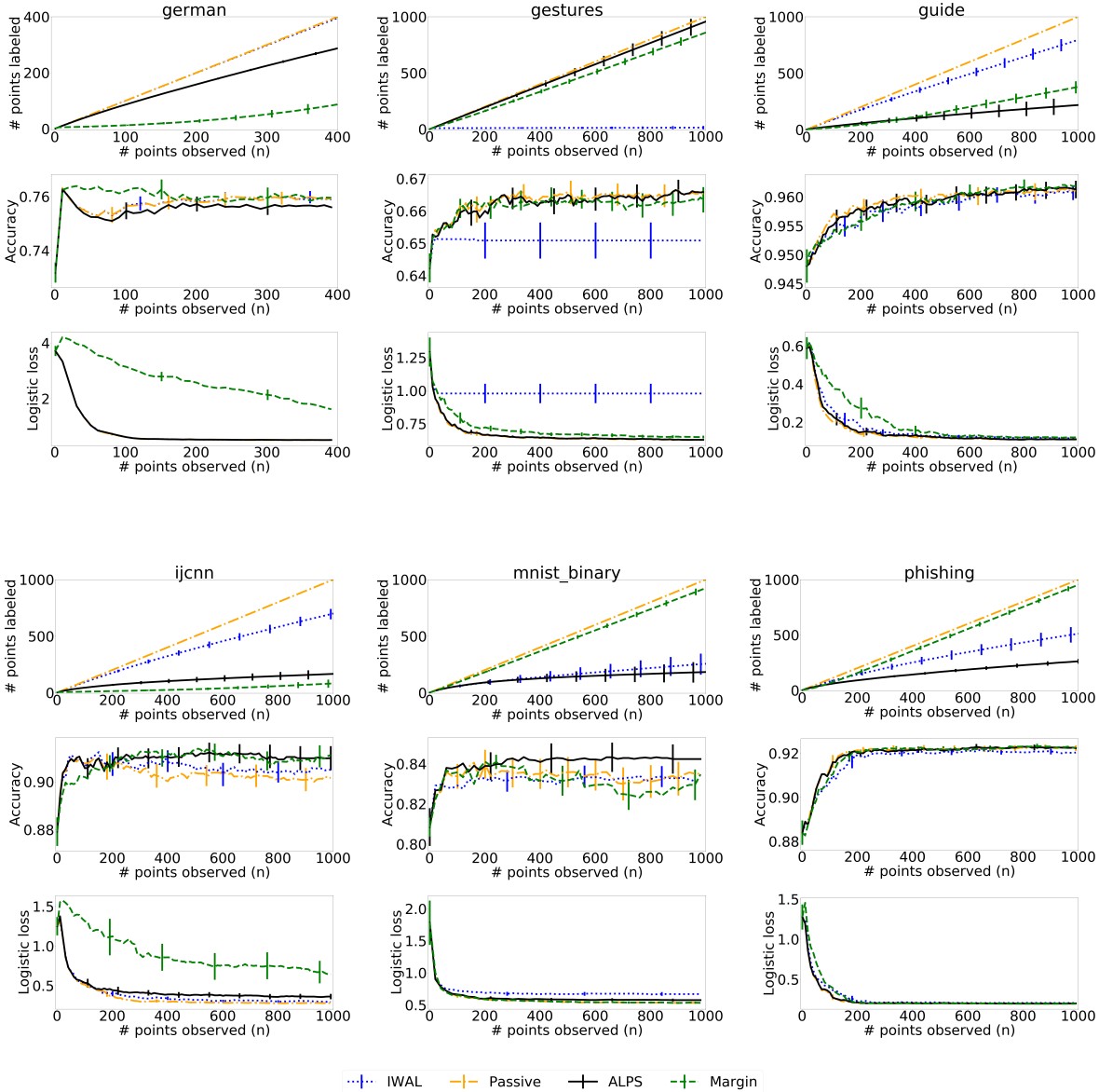

Figure 10: Comparison of the ALPS algorithm, the IWAL algorithm, the ex-post best margin algorithm and the passive learning algorithm for 4-layer networks (2/3). The figures shows the mean over the 50 trials and twice its standard error of the number of labeled points, accuracy and logistic loss on the test set of the model returned by each algorithm as a function of the number of observed points (or rounds) $n$.

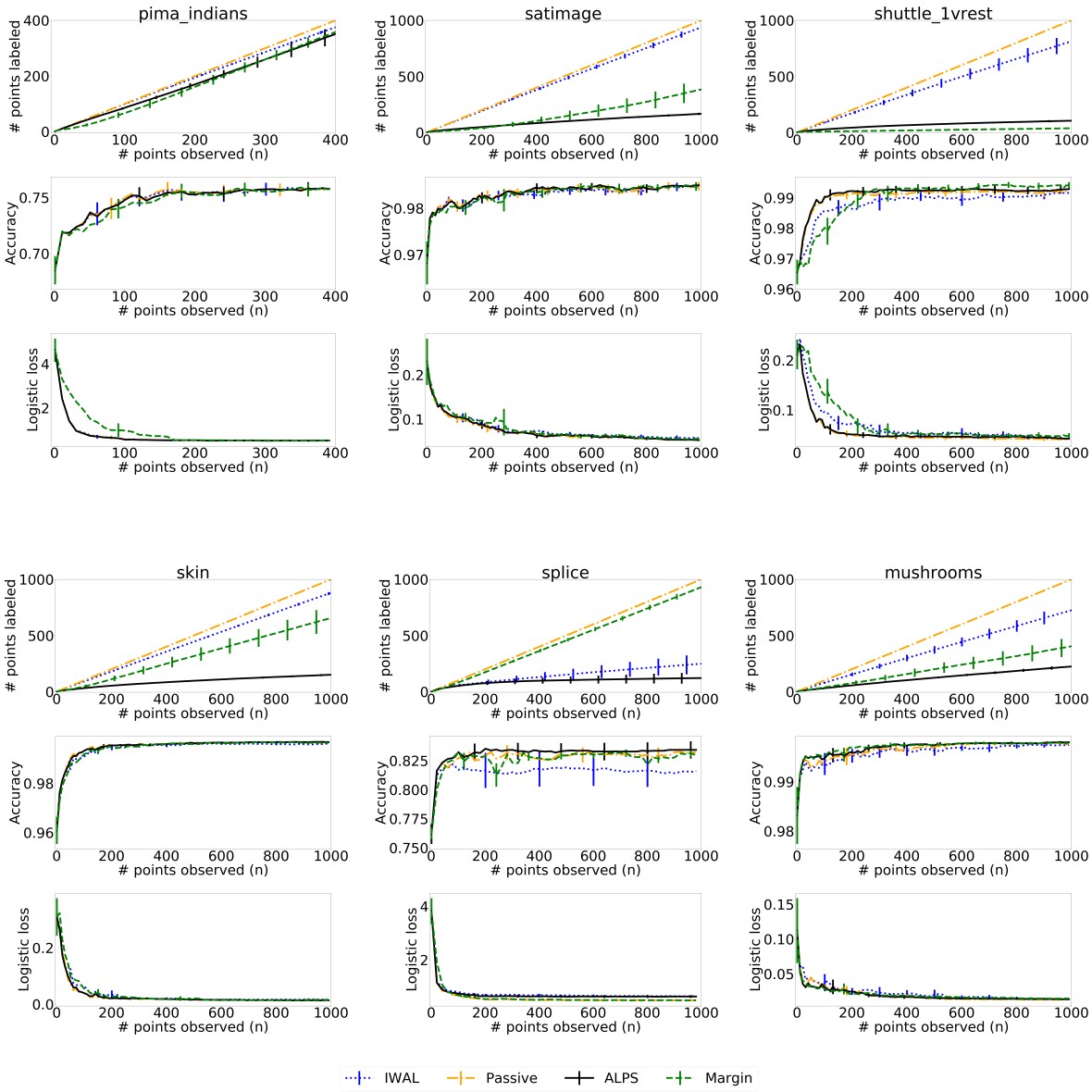

Figure 11: Comparison of the ALPS algorithm, the IWAL algorithm, the ex-post best margin algorithm and the passive learning algorithm for 4-layer networks (3/3). The figures shows the mean over the 50 trials and twice its standard error of the number of labeled points, accuracy and logistic loss on the test set of the model returned by each algorithm as a function of the number of observed points (or rounds) $n$.

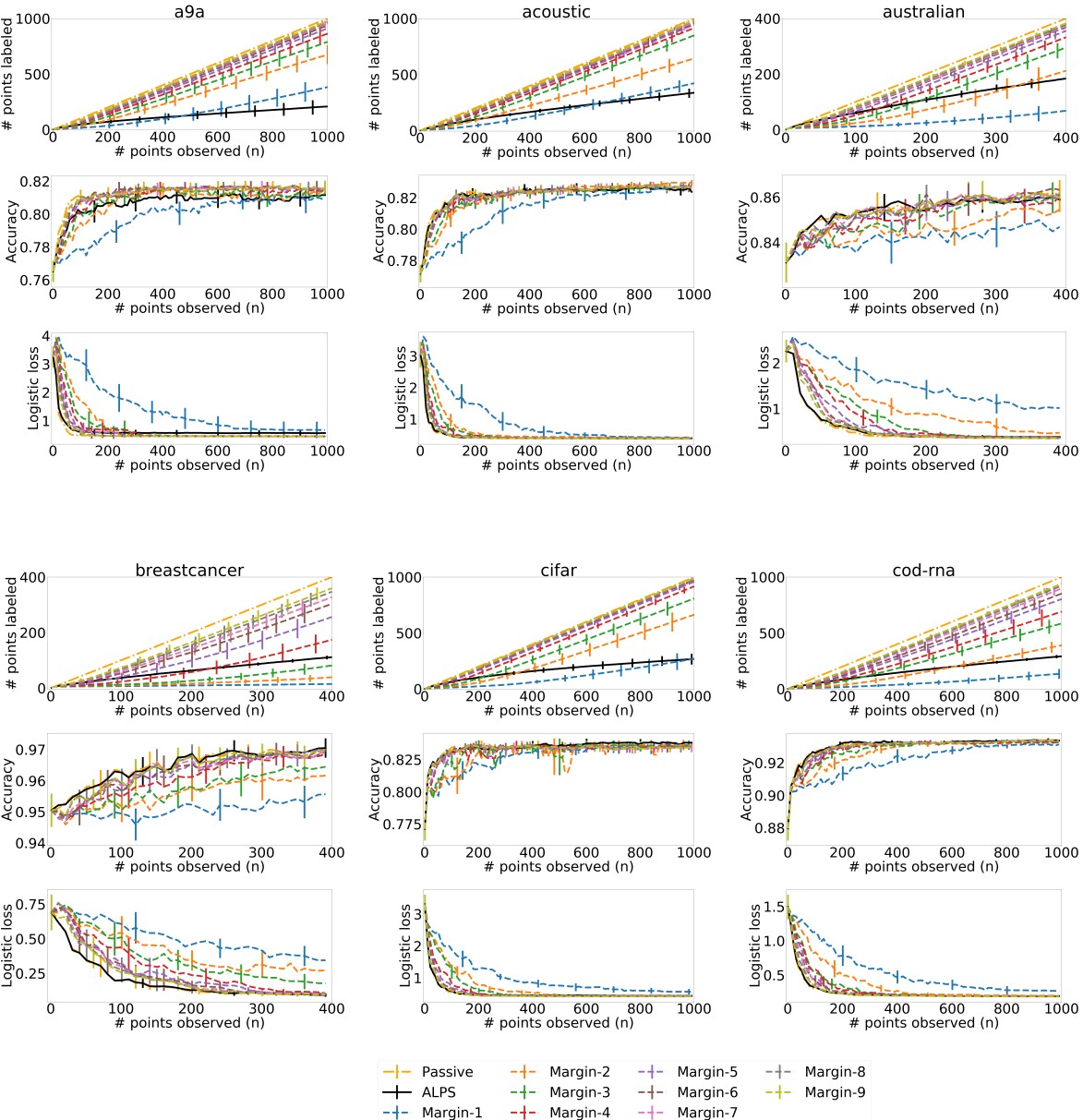

Figure 12: Comparison of all nine margin algorithms to passive learning and ALPS for 4-layer networks (1/3). The margin algorithms are ordered from the smallest to the largest threshold in $\bar{\Gamma}_m$, where Margin-1 corresponds to the smallest threshold, Margin-2 to the second smallest, etc. The figures shows the mean over the 50 trials and twice its standard error of the number of labeled points, accuracy and logistic loss on the test set of the model returned by each algorithm as a function of the number of observed points (or rounds) $n$.

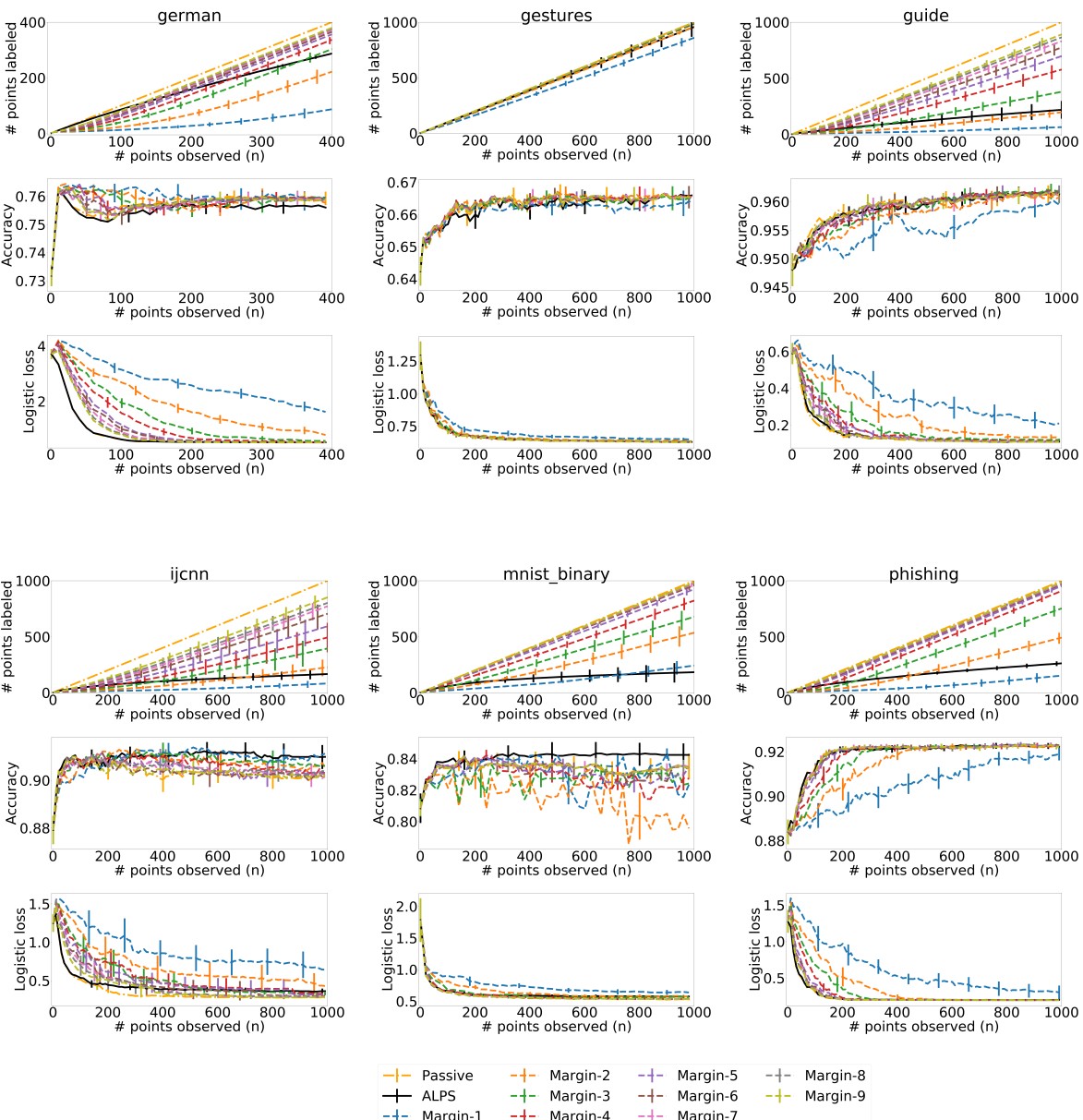

Figure 13: Comparison of all nine margin algorithms to passive learning and ALPS for 4-layer networks (2/3). The margin algorithms are ordered from the smallest to the largest threshold in $\bar{\Gamma}_m$, where Margin-1 corresponds to the smallest threshold, Margin-2 to the second smallest, etc. The figures shows the mean over the 50 trials and twice its standard error of the number of labeled points, accuracy and logistic loss on the test set of the model returned by each algorithm as a function of the number of observed points (or rounds) $n$.

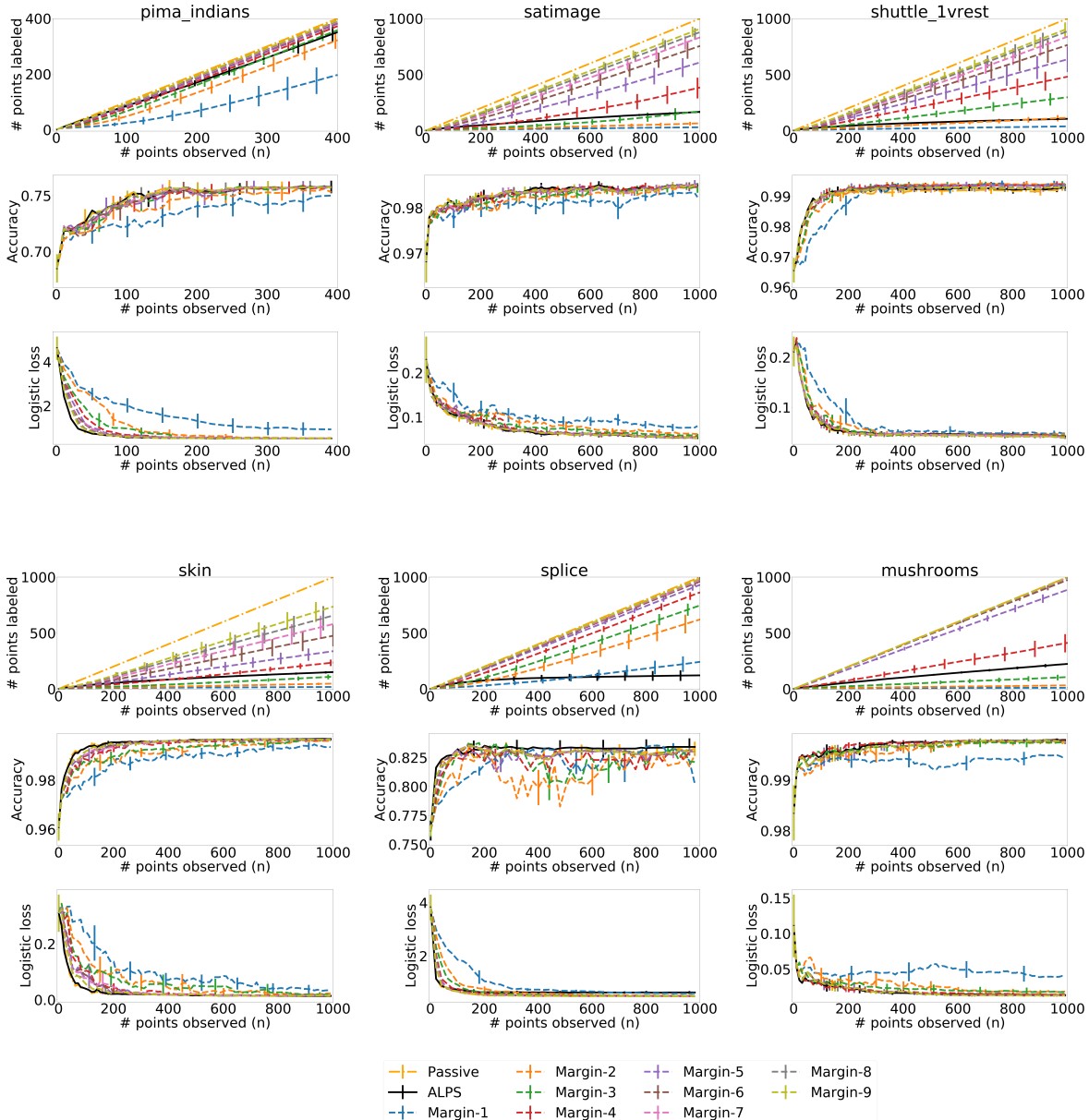

Figure 14: Comparison of all nine margin algorithms to passive learning and ALPS for 4-layer networks (3/3). The margin algorithms are ordered from the smallest to the largest threshold in $\Gamma_m$, where Margin-1 corresponds to the smallest threshold, Margin-2 to the second smallest, etc. The figures shows the mean over the 50 trials and twice its standard error of the number of labeled points, accuracy and logistic loss on the test set of the model returned by each algorithm as a function of the number of observed points (or rounds) $n$.