# OpenReview forum: "Online Active Learning with Surrogate Loss Functions"
_NeurIPS.cc/2021/Conference — NeurIPS 2021 Spotlight_

### Official Review · Reviewer_AgoF · 2021-07-13

**Rating:** 7
**Confidence:** 3

**Summary:**

This paper presents Actively Learning over Pseudo-Labels for Surrogate Losses (ALPS), an active learning algorithm for binary classification with surrogate losses in the streaming setting. The main idea is to generate pseudo-labels for points whose labels are not requested and train on the union of data points with ground-truth labels and those with pseudo-labels. Since the pseudo-labels may not be accurate, the authors propose requestor functions that have a final say in whether a pseudo-label is used, and in effect, enable control over the noise generated by erroneous pseudo-labeling. Under an assumption involving the requestor function class, the authors prove bounds on the generalization error and label complexity of ALPS that are roughly on the same order of prior work (e.g., IWAL).

**Limitations And Societal Impact:**

Yes

**Main Review:**

Strengths

- The paper is generally very well-written and organized, with a clear exposition.

- Good coverage of related work (Sec. 1) and comparison of the proposed method to prior approaches. I also appreciate the discussion concerning the reduction of ALPS to DHM under certain conditions (Lines 107-113), which to me highlights the generality of the approach.

- I found the ideas of requestor functions and the definition of the version space as a function of both the requestor function and hypothesis class to be interesting and novel. The authors were also forthright about the relation of requestor functions to similar ideas like abstention functions from prior work.

- At a high level, the theoretical analysis of the generalization error and label complexity seems sound to me. I also think the relaxed assumption (Assumption 2) is relatively mild in the context of, for example, deep learning due to the high-confidence predictions that are typically produced.

- The authors present compelling empirical evaluations on a wide variety of data sets and compare to IWAL and instantiations of a margin-based algorithm.


Weaknesses

- Relation to heuristic approaches: given that the primary appeal of the algorithm seems to be its practical effectiveness compared to prior work, e.g., IWAL (which has roughly the same guarantees), can anything be said about the performance of ALPS relative to SOTA active learning approaches like BADGE (which is actually cited in the paper as [1])? Although experimental comparisons would be best (the code for BADGE is publicly available [1]), even a qualitative discussion would be insightful.

- The method seems to work only for binary classification tasks. Can it be extended to multi-class classification?

- This is mostly a stylistic concern, but the introduction is quite long and the full extent of the contributions of the paper are not made clear until (roughly) page 3. I wonder whether partitioning the introduction section into two sections (intro and related work) or adding at least one subsection in the introduction for related work would improve the exposition.


[1] https://github.com/JordanAsh/badge

**Time Spent Reviewing:**

7

---

> ### Author Response · Authors · 2021-08-09
> **Response to Reviewer AgoF**
>
> $\textbf{On the comparison to SOTA methods like BADGE.}$
>
> We agree that this would be an interesting comparison, but it is not an easy one to attempt since many of these algorithms are not designed to work in a streaming setting. For example, one of the key steps in BADGE leverages the entire unlalled pool to select examples by running kmeans++ seeding.
>
> Qualitatively,  BADGE and ALPS differ in the way they deal with the bias loss estimates induced by the active selection of examples.  Specifically, the  sample selected by BADGE will be biased with respect to marginal distribution on the instance space X, as it selects sets of diverse low confidence examples. On the other hand, ALPS does not bias the marginal distribution on X, and instead uses pseudo-labels, thereby inserting a controlled bias in the label space Y. At the same time, when ALPS is run with margin requester functions, ALPS is akin to BADGE in that both focus on requesting the labels of low confidence examples.
>
> $\textbf{On the extension to multi-class classification.}$
>
> Despite the fact that we did not explicitly investigate it, we strongly believe that it is possible to extend our algorithm and associated analysis to multi-class classification.
>
> $\textbf{On the lengthy introduction.}$
>
> Thanks for this comment. We will partition the introduction to improve exposition.

---

> > ### Comment · Reviewer_AgoF · 2021-08-24
> > **Response to authors**
> >
> > Thank you for your response.
> >
> > I think comparisons with BADGE -- even in the offline setting -- would make the paper more compelling. The same could be said of the multi-class classification extension and evaluations in this setting. I understand, however, that these may be avenues for future research. Hence, I decided to keep my score unchanged.

---

### Official Review · Reviewer_nx35 · 2021-07-16

**Rating:** 6
**Confidence:** 3

**Summary:**

This paper studies the problem of online active learning for classification. In order to learn a classification algorithm, the paper proposes to employ surrogate loss function. The error of the proposed method is analyzed and some empirical results are presented.

**Limitations And Societal Impact:**

The authors discuss some of limitations in their work. Also they can add that their proposed algorithm needs storing observed data samples to make decision.

**Main Review:**

The paper focuses on a problem that has been studied extensively. However, I think the contribution of this work relevant to prior works is not clear. Firstly, the authors does not explain that whether considering surrogate loss function results in obtaining tighter bound for error or not. In fact, the paper makes two additional assumptions compared to existing algorithms such as DHM (as it is called in the paper). Thus, it is expected that under these two assumptions  a tighter regret can be obtained. Otherwise, the error bound of the proposed algorithm is not tighter than existing works under two additional assumptions. This means that the paper does not make a significant contribution compared with existing works. Therefore, more discussion on error bounds and comparison with error bounds of other algorithms is required. Also there are two minor comments. 1- It seems that the proposed algorithm needs to store previously observed data samples while one of the reasons that online learning algorithms may be used is that data cannot be stored in batch. Thus, the proposed algorithm cannot be leveraged in these cases. Hence, this can be account as a limitation of the proposed algorithm. 2- In the theorem 1, it is stated that the inequality holds with probability at least $1-\delta$. However, there is not any $\delta$ in the inequality. It is highly recommended presenting the upper bound of error in theorem 1 using some terms which include $\delta$.

==============Review Update===============

Thanks the authors for the rebuttal. The authors' responses addressed most of my concerns. I think revisions that authors want to make will improve the paper. I am happy to increase my score.

**Time Spent Reviewing:**

5 hours

---

> ### Author Response · Authors · 2021-08-09
> **Response to Reviewer nx35**
>
>
> $\textbf{On 0-1 losses, surrogate losses, and contribution of the paper.}$
>
> Perhaps, there might be a misunderstanding here. The goal of this paper is $\textit{not}$ to achieve improved guarantees for the 0-1 loss by leveraging a surrogate loss. Instead, we aim to study (both theoretically and experimentally) the binary classification problem in terms of the surrogate loss itself. Surrogate losses, like the logistic loss, are used in training to solve tractable optimization problems, $\textit{but also for evaluation}$, as they result in more calibrated metrics.  Please recall lines 18 - 32. Note that previous works also focus on deriving guarantees with respect to surrogate losses (e.g. see the theoretical analysis in [7] of IWAL, which is an important active learning algorithm.)
>
> In the recent active learning literature, the gap between theory and practice has been widening at a faster rate. We thus try to bridge this discontent by deriving an algorithm that not only works well in practice, but also admits solid theoretical guarantees. In particular, we are interested in deriving an algorithm that is closer in spirit to margin-based algorithms (such as uncertainty sampling), which are widely adopted in practice but lack full theoretical understanding in the general agnostic setting. Please consider more carefully the connection between the theory sections of our paper and its experimental counterpart (Section 5). There, we show that ALPS experimentally overcomes the best post hoc tuned margin algorithm, as well as other baselines, like IWAL.
>
> In retrospect, we think that our usage of terms like ``surrogate loss” might have been a bit  misleading, and so might have been our sentence in lines 251-253. We will clarify this in the paper.
>
> $\textbf{On improved bounds and comparison to IWAL [7].}$
>
> The bounds of the IWAL algorithm in [7] are already unimprovable in the agnostic setting since the lower bound in [7] shows that the risk bounds of IWAL and ALPS are effectively tight.  Please recall the paragraph after Theorem 2 and lines 303-304. At the same time, it is well known (and further confirmed by our experiments) that IWAL does not work well in practice. In contrast, our algorithm admits tight theoretical guarantees in general scenarios and is also performing well in experiments.
>
> $\textbf{On the storage of past examples.}$
>
> The reviewer is right, our algorithm (as other algorithms like DHM) needs to store previously observed data. We will make this more clear in the text.
>
> $\textbf{Delta in theorem 1}$
>
> The delta is hidden in the $\tilde{O}$ notation, which we state in the line right above Theorem 1. We will be happy to expose the $\delta$ in the theorems for more clarity.

---

### Official Review · Reviewer_VTZ8 · 2021-07-16

**Rating:** 7
**Confidence:** 4

**Summary:**

The authors propose an active learning algorithm in the streaming setting for binary classification tasks. The algorithm leverages weak labels to minimize the number of label requests and trains a model to optimize a surrogate loss on a resulting set of labeled and weak-labeled points. The paper contains a number of theoretical results and supporting empirical results. Overall, its a very strong paper.

**Limitations And Societal Impact:**

The authors state their work does not have any negative societal impact. The limitations discussed are mainly in the theory. I think another point worth discussing is how this approach can be used in a batch setting and for DNNs.

**Main Review:**

Pros of this paper:
- Has theoretical guarantees in addition to good empirical performance. Theoretical guarantees hold in the agnostic case where the true labeling function is not realizable.
- Furthermore, theoretical guarantees have generalization complexity and LABEL complexity
- Has good discussion on existing work
- Aside from its length, the introduction is very, very good. It is well-written, explains detriments in previous theoretical results in active learning, and clearly qualifies what setting it is tackling and how its solution is effective.
- Uses a very large assortment of real-world datasets
- Notations are good; the definition of consistency matches the usual definition
- The explanation for ALPS naturally develops itself, making it easier to follow along.
- Theoretical findings have bounds that match proven lower bounds up to log terms, showing near optimality.

Cons of the paper/questions:
- Models used are neural networks, but they are not deep networks (they are MLPs).
- Will the techniques work for DNNs? If yes, how and what needs to change? Will be good to add some intuition for practitioners particularly if this needs to be used in a batch setting
- Introduction is a bit lengthy; perhaps it could be broken down into more sections
- Link in Algorithm 1 (superscript 2) is broken; doesn't tab to footnote
- Alg 1 notation for h_y could be refined to show that h_{-1} and h_{1} are both being trained as mentioned in lines 164-173
- ALPS depends on the existence of consistent hypotheses on the relevant subsets of the data, which might not always be the case.
- Line 203: Is the notation used for list?

Summary:
Largely, I think this paper is very well written in that it develops the delivery of its results very well in the introduction and that it explains the workings of the algorithm in a logical order. They provide an astounding amount of experiments/datasets. Hence, I would recommend that this paper be accepted for these reasons, on top of the fact that it provides the empirical edge to other similar AL works for online binary classification.

**Time Spent Reviewing:**

3 - 4 hours

---

> ### Author Response · Authors · 2021-08-09
> **Response to Reviewer VTZ8**
>
> $\textbf{On the batch setting and usage of DNNs.}$
>
> In the batch setting, the active learner must select a set of B points to be sent for labeling as opposed to one point at a time. In this setting, we can consider two frameworks: online and pool.
>
> For the online framework, a recent paper called “Understanding the Effects of Batching in Online Active Learning” by Kamin et al. AISTATS 2020 presents a general meta-algorithm that extends a streaming algorithm to the batch setting. Effectively, the streaming algorithm’s internal state is frozen until a batch of B points is selected.  We have verified that the same technique can be directly applied to the ALPS algorithm and its theoretical guarantee can be extended to batch settings with only a mild dependency on the batch size.
>
> For the pool framework, one could simply stream the data and use the above online batch solution. On the other hand, we would not be leveraging the entire unlabeled pool at our disposal. Finding a solution that uses the entire unlabeled pool is part of our next steps for this project.
>
> Since we ran extensive experiments on 18 datasets, we used smaller networks, but nothing
> prevents us in principle from running DNNs. Due to the longer training and inference time of DNNs, our suggestion in this case is running this algorithm in the batch setting described above.  We will add comments along this line to the experimental section of the paper.
>
>
> $\textbf{ ``ALPS depends on the existence of a consistent hypothesis … which might not always be the case”.}$
>
> This existence  is ensured $\textit{by construction}$. Recall that consistency of the hypothesis is always with respect to the pseudo-labeled set. ALPS will only pseudo-label a point to, say, +1 if
> ${\hat err_n}(h_{+1}) -  {\hat err_n}(h_{-1})  > {\tilde \Delta_n}$
> or if $h_{-1}$ does not exist. See lines 170-173. That is, a consistent hypothesis with +1 label must exist for this point to be added to the pseudo-label set. Otherwise, we ask for the label of this point. We will explain this more carefully in the main text.
>
> $\textbf{Other comments.}$
>
> Thank you for the notational suggestions. Line 203 is a notation to denote a list. We will clarify this in the text and we will certainly break the introduction into more sections.

---

### Official Review · Reviewer_TknT · 2021-08-31

**Rating:** 6
**Confidence:** 4

**Summary:**

This paper focuses on stream-based (online) active learning for training classifiers in the binary classification setting. In particular, the authors propose a strategy, the so-called ALPS*, that trains classifiers by optimizing a surrogate loss over both the labeled instances and sequentially predicted data points by the learner. Theoretical guarantees on label complexity and error of the returned model are provided, and are shown to be competitive with the existing bounds. A set of experiments are conducted on datasets using multi-layer perceptrons as classification models, where the performance of ALPS is illustrated to have good performance with small labeling cost.


*Actively Learning over Pseudo-labels for Surrogate losses

**Ethical Concerns:**

To my assessment, I do not see any ethical concerns within this work.

**Limitations And Societal Impact:**

No discussion of societal impact, limitations on the theoretical assumptions have been addressed.

**Main Review:**

-This work uses some existing techniques such as importance weighting, margin based functions in their building of final algorithm. The intuition is provided to understand the role of each component, for instance, how importance weighting helps with reducing the noise in hypothetical labels through requester functions. It is also mentioned by the authors in the paper that the proposed method is an extension of [1] to surrogate loss functions. With differences identified, the final methodology is sufficiently original in my assessment.

-The theoretical results on label complexity as well as the excessive risk of returned hypothesis (over the risk of best one in the hypothesis class) are competitive, although one may argue with certain assumptions at times such as access to a large unlabeled data pool (as in, once it is the case, there already exist several competitive pool-based methods over the online approaches, which makes prediction on the pool and measure disagreement among all pool examples). Overall, together with its empirical performance, the contributions seem adequately significant.

-The paper is well written and clear.

-My main argument would be the experiments. Namely,\
--I find the experimental setting limited. That is, if the focus is active learning hence reducing the number of label requests, then I believe a better comparison would be where each model is restricted to the same labeling budget throughout the entire stream. For instance, the query probability in the IWAL approach can be linearly scaled with a constant such that it queries the same amount of labels with ALPS by the time n rounds end, then illustrating the performance of methods over the rounds would be very informative. Often times, performance may change only a little after a certain amount of requests, hence it could help to see how they compare when they are restricted to limited amount of queries. As a note, I think passive learning in active learning literature refers to _random sampling_ than querying all labels (in this paper, it would correspond to $p_n$ being fixed at each round where $Q_n\sim Ber(p_n)$) throughout the stream, similar to label efficient prediction in online learning. As trivial as it may sound, the passive learning is shown to be very robust in the existence of noise studied in the paper. I recommend conducting experiments in which proposed approach is compared to the passive learning baseline in terms of its robustness.\
--The model space could be extended further than multi layer perceptrons.\
--There are other approaches that can be adapted/designed-for online setting [2,3] that uses disagreement among the classifiers (through measures like entropy). It would have been informative to see how ALPS performs compared to those approaches that are generally good.



[1] S. Dasgupta, D. Hsu, and C. Monteleoni. A general agnostic active learning algorithm. In Proceedings of NeurIPS, pages 353—360, 2008\
[2] Ido Dagan and Sean P. Engelson. Committee-based sampling for training probabilistic classifiers. In Machine Learning Proceedings 1995, pages 150–157. Elsevier, 1995\
[3] Burr Settles. Active learning literature survey. Technical report, University of Wisconsin-Madison Department of Computer Sciences, 2009.

**Time Spent Reviewing:**

3

---

> ### Author Response · Authors · 2021-09-01
> **Response to Reviewer TknT**
>
> We thank the reviewer for the detailed feedback on our paper.
>
> $\textbf{On the access to a large unlabeled pool of data.}$
>
> This assumption is only aimed at making our analysis simpler, but is not strictly needed (please recall Footnote 4). In the theoretical section (see line 225), we simply discuss how one $\textit{could}$ use an unlabeled pool $\textit{if available}$ in order to construct a convenient set of requester functions. However, in our experiments we do not use an unlabeled pool to generate requester functions or to estimate the requesting rate. Instead, we define the set of requester functions to be margin-based functions with varying thresholds, and then estimate the requesting rate based on the data processed thus far. We will better clarify in the paper.
>
> $\textbf{On the reported experimental results.}$
>
> Even though the comparison suggested by the reviewer is natural, notice that enforcing extra constraints on algorithms that don’t naturally admit them, may hurt their performance, thereby resulting in a potentially unfair comparison. In particular, changing the number of requests made by an active learning algorithm directly affects its generalization. Thus, in our empirical comparison, we allow each algorithm to decide how often to query.
>
> As an example, consider the margin sampling algorithm. In order for margin sampling to match the number of requests made by ALPS, one needs to vary its threshold $\gamma$. However, picking such a threshold could result in either 1) a worse generalization or 2) a worse label complexity.
>
> 1) Worse generalization example: In Figure 6 (see Sect. D in the appendix), the australian plot (top right) shows that if we pick a margin threshold (orange line) that matches the number of labels requested  by ALPS (black line) in the “# of points labeled” figure, then the generalization measured by either  accuracy (second from top) or logistic loss (third from top) is considerably worse (compare orange to black line). That is, the generalization of the margin algorithm appears considerably worse than it could be if a slightly larger threshold had been chosen.
>
> 2) Worse label complexity example: In Figure 7 (Sect. D in the appendix), the german plot (top left) shows that margin with the smallest threshold (blue line) achieves the same accuracy as ALPS while using fewer labels. If instead we force margin to request a similar number of labels as ALPS (orange line), then it would make margin sampling appear to admit the same accuracy as ALPS with the same number of labels. That is, we could incorrectly conclude that margin and ALPS admit the same performance on this dataset.
>
> The above also clearly holds if we linearly scale IWAL’s query probability by some constant $c$. If the scalar $c$ is greater than 1, then IWAL requests more labels than it would need to achieve the same generalization guarantee. If the scalar $c$ is less than 1, then IWAL’s generalization guarantee is worsened.
>
> Note that the comparison that the reviewer is advocating for can essentially be carried out for margin sampling since our paper presents results for several thresholds. In this case, we again find that ALPS outperforms margin sampling for most datasets. For IWAL, we would have to re-run the algorithm to do this comparison. Yet, perhaps this experiment might not be necessary, since it is known in practice that IWAL underperforms compared to margin sampling. (Recall that we included IWAL in our experiments since, unlike margin sampling, IWAL admits theoretical guarantees – see first paragraph of section 5 and also line 305-306).
>
> $\textbf{On passive learning and random sampling.}$
>
> The reviewer may want to observe that our passive learning algorithm, and the way we report performance are $equivalent$ to what they are suggesting about random sampling. Note that in our experimental setting the training set is randomly shuffled. Hence running passive learning up to round $m*p$, where $m$ is the size of the training set, and then stopping to measure performance on the test set is statistically equivalent to a random sampling algorithm that sweeps over the entire training set, and queries labels with independent probability $p$. This equivalence can be made more formal, but is essentially due to the fact that all models we train do not change their state unless more labels are acquired. So, for instance, if the training set size is $m=1000$,  we expect the performance of a random sampler with $p = 0.2$ to correspond to the performance of passive learning that has been tested at round $200$. We will mention this equivalence in the revised version of the paper.
>
> $\textbf{On extending the model space.}$
>
> Yes, we agree that a natural next set of experiments would be to move beyond multi-layer perceptrons.
>
> $\textbf{ On comparing to other approaches.}$
>
> Out of the algorithms in survey [3], we decided to test against margin sampling since recent active learning empirical studies have shown that this algorithm outperforms many state-of-the-art active learning algorithms, see e.g., references [47, 12] in our paper. Notice in particular that in [12], the study tests both entropy-based algorithms and committee-based algorithms on 60+ datasets.  Please also see the discussion starting at line 57 in our paper.

---

> > ### Comment · Reviewer_TknT · 2021-09-03
> > **Post-rebuttal**
> >
> > Thanks to the authors for their prompt reply and efforts to explain further details!
> >
> > -- I am aware of that adding an extra constraint on the labeling budget will hurt the performance of algorithms, but I disagree with it being unfair if one wants to compare them for a given budget. It is more of a practical concern given motivation is to reduce the number of requests. I also acknowledge that in online learning papers, this type of comparison in the paper is favored, yet I still value comparisons based on a fixed labeling budget for practical reasons.
> >
> > -- Naturally, quitting querying for passive learning during streaming will result in random sampling due to random shuffling, but we do not have this result in the paper, which is what I have been referring to. I think it is a must to do that for random sampling for the sake of completeness (even though the datasets in consideration might not show the robustness of it).

---

> > > ### Author Response · Authors · 2021-09-03
> > > **Response to Reviewer TknT**
> > >
> > > We thank the reviewer for kindly taking the time to read and respond to our comments.
> > >
> > > $\textbf{On comparing algorithms with a fixed budget.}$
> > >
> > > We will add to our empirical results the comparison based on a fixed labeling budget. Please observe that, when focusing on ALPS vs. margin sampling, the comparison the reviewer is advocating for is effectively there (though, perhaps, not as explicit as it could have been). Varying the threshold in margin sampling is equivalent to forcing a label request budget. Under this comparative scheme, we again find that ALPS outperforms margin sampling.
> > >
> > > We also expect that ALPS will outperform IWAL under the fixed labeling budget comparison, since in all experiments we are aware of, IWAL considerably underperforms compared to margin sampling, and this is also reaffirmed by our own experiments.
> > >
> > > $\textbf{On comparing to random sampling with a constant label rate.}$
> > >
> > > Perhaps there is a small misunderstanding here, but when we say “passive” in the plots, it seems to us that we have exactly what the reviewer has in mind.  Each reported value of our passive learning curve at each round $n$ in Figure 2 is equivalent to quitting querying for passive learning at round $n$ as the data is streamed in a random order.
> > >
> > > So, for instance, at round $n$=200 of the passive curve (orange line) for the ijcnn dataset in Figure 2, we effectively randomly sample and reveal the label of 200 points out of the training set, choose a model that corresponds to an ERM over these 200 points, and evaluate this model’s accuracy on a test set. The accuracy of this model at $n$=200 is around 0.92, which we show in the second row of the plots in Figure 2.
> > >
> > > We will make sure to clarify this in the paper, and we invite the reviewer to respond to this comment if they feel we have misunderstood their point.

---

> > > > ### Comment · Reviewer_TknT · 2021-09-04
> > > > **Post-rebuttal**
> > > >
> > > > Thank you for further clarifications!

---

### Decision · Program_Chairs · 2021-09-27

**Decision:**

Accept (Spotlight)

**Comment:**

This paper proposes a fresh approach to active learning with non zero-one losses. The proposed algorithm has theoretical guarantees that are competitive with existing works, while also performing in practice better than existing practical techniques, in a large set of experiments. This double advantage is of significant value to the advancement of the field. Moreover, the approach includes fresh ideas that could open up new avenues of investigation. Some reviewers have suggested additional experiments. We encourage the authors to add those to their final version.